# 🤗 HUGGING CARBON: QUANTIFYING THE TRAINING CARBON EMISSIONS OF AI MODELS AT SCALE

## ABSTRACT

The scaling-law era has propelled artificial intelligence (AI) from research into a global industry, but its rapid growth raises concerns over energy demand, carbon emissions, and environmental sustainability. Unlike traditional sectors, AI still lacks systematic methodologies for comprehensive carbon accounting, leaving open the questions of how large the problem is today and how large it might be in the near future. We propose a FLOPs-based framework to estimate training emissions of open-source models on Hugging Face, introducing a tiered approach to handle uneven disclosure quality. Compute is converted to energy using hardware efficiency characteristics and then to emissions using the carbon intensity of the relevant grid, which we summarize as an AI Training Carbon Intensity (ATCI, emissions per compute) and for which we report an empirical reference value to enable quick model-level estimates. Our results show that training the most popular 5,234 models (with over 5,000 downloads) emitted approximately $5.8 \times 10^4$ tons of carbon emissions. These findings provide the comprehensive industry-scale estimate of AI's training footprint and a practical methodology to guide future standards and sustainability strategies.

## 1 INTRODUCTION

In the scaling-law era, artificial intelligence (AI) has expanded from academic research into an industry worth hundreds of billions of dollars today, and is projected to reach several trillion dollars by 2030 (UNCTAD, 2023). Large models, spanning computer vision (CV) and large language models (LLMs), are now deployed across critical fields such as robotics, the Internet, energy, and industrial sectors. This rapid scaling of model size, data, and parameters is driving unprecedented demands for energy (IEA, 2024; Strubell et al., 2020), water (Li et al., 2023; Morrison et al., 2025), and materials (Lee et al., 2025). Concerns over AI's environmental sustainability are intensifying (Schwartz et al., 2020; Wu et al., 2022; Bashir et al., 2024), as rising carbon emissions risk accelerating climate change and resource strain.

However, these concerns often remain conceptual. While policymakers and researchers broadly acknowledge the challenge, there is still a lack of systematic estimates to the questions of **"how large is the problem today"** and **"how large might it in the near future"**. In contrast, traditional industries, such as manufacturing and agriculture, already follow established methodologies (Eggleston et al., 2006; IPCC, 2014) and disclosure standards (ISO, 2018) for product-level life-cycle footprints (Bhatia et al., 2011; myclimate, 2023) as well as industry-wide carbon accounting (IPCC, 2022). AI, despite its widely recognized environmental implications, still lacks consistent reporting and scalable methodologies for estimating training emissions across a wide range of model families and modalities. Comprehensive and long-term disclosure of the environmental costs of model development and deployment remains highly limited, and the quality of existing disclosures is often inadequate. This gap makes even a basic understanding of AI's current environmental impacts a pressing and unresolved challenge.

Here, we make a further attempt to bridge these gaps. Unlike previous studies that focused primarily on the carbon footprint of individual models (Strubell et al., 2020; Morrison et al., 2025), we aim to provide a broader, industry-scale perspective on AI's emissions by offering a conceptual estimate of its overall impact. As a lens for this investigation, we examine open-source models hosted on Hugging Face (HF), the most widely used repository and distribution platform for AI

models. The models available on Hugging Face represent a substantial share of the open-source community's collective efforts, making them a valuable proxy for estimating emissions in practice. By accounting for the training emissions of these models, we seek to shed light on how much carbon AI model training has already emitted and how much additional emission its continued scaling may generate. Given the limited quality and scope of existing disclosures, our goal is not to provide fully accurate numbers but to develop an accounting framework supported by large-scale estimation and cross-validation. We hope this framework can offer a meaningful bigger-picture view of AI's environmental impact, both at the model level and across the industry.

Accounting for the training carbon footprint of models hosted on Hugging Face is far from straightforward, requiring a practical methodology. Although open-source models provide a relatively transparent basis for analysis, their disclosure quality remains uneven, with many fields requiring manual completion or inference. Reproducing the training process for millions of models would be both infeasible and environmentally wasteful. To address this, we introduce a FLOPs-based estimation framework. The key idea is to first approximate the total computational cost (in FLOPs) required to train a given model. This quantity is then converted into energy consumption based on the efficiency characteristics of the hardware likely used for training, and finally into carbon emissions by applying the carbon intensity of electricity in the relevant region. This conversion can be interpreted as assigning an AI training carbon intensity (ATCI, training emissions per compute), which reflects both hardware energy efficiency and regional energy mix. We further provide an empirical reference value for this intensity, offering a practical baseline for subsequent studies and enabling quick estimation of model-level training emissions.

In practice, we begin by focusing on models with high download counts and wide adoption, as they not only exert greater influence but are also more likely to provide at least partial transparency regarding their training. Based on the completeness of disclosed information, we classify these models into three tiers: Tier 1 models disclose sufficiently detailed information, allowing us to cross-check their carbon emissions from multiple perspectives; Tier 2 models have partial gaps in disclosure, but these can be reasonably inferred using the data accumulated from Tier 1; and Tier 3 models disclose very limited or no usable information, requiring us to rely on empirical assumptions for rough estimation. This tiered categorization enables our framework to remain systematic and applicable despite substantial heterogeneity in disclosure practices.

Our estimates suggest that training the 5,234 models with more than 5,000 downloads produced approximately **58,000 tons of $CO_2$e**. As shown in Figure 1, the total footprint is comparable to about 1.5% of the passenger-car emissions of a mid-sized European country. The number of models on Hugging Face continues to rise annually as thousands of new popular models are released each year, underscoring its non-negligible emission scale within the open model ecosystem.

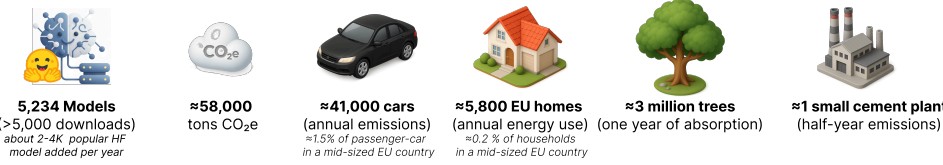

| 5,234 Models | ≈58,000 | ≈41,000 cars | ≈5,800 EU homes | ≈3 million trees | ≈1 small cement plant |
| (>5,000 downloads) | tons $CO_2$e | (annual emissions) | (annual energy use) | (one year of absorption) | (half-year emissions) |
| *about 2-4K popular HF model added per year* | | *≈1.5% of passenger-car in a mid-sized EU country* | *≈0.2 % of households in a mid-sized EU country* | | |

Figure 1: Estimated training emissions from 5,234 Hugging Face models, compared with equivalent real-world scales (cars (Tiegte et al., 2021), homes (Eurostat, 2025), cement plants (IEA, 2025), and trees (Franklin Jr & Pindyck, 2024); tree absorption = 18 kg $CO_2$/tree/year).

## 2 RELATED WORK

**Sustainability of AI** requires quantifying and mitigating the environmental costs of developing and deploying AI models. Early awareness came from work on energy and policy considerations in deep learning: Strubell et al. quantified the carbon emissions of training large neural networks and argued that computing should be treated as a scarce resource (Strubell et al., 2019; 2020), while Schwartz et al. (2020) proposed the "Green AI" agenda, calling for efficiency and environmental impact to be considered alongside accuracy. Patterson et al. (2021) later estimated emissions from models such as GPT-3, showing how data-center efficiency and energy mix strongly affect outcomes.

Subsequent research broadened the scope beyond individual case studies. Wu et al. (2022) surveyed the environmental impacts of AI across data, algorithms, and hardware, and Dodge et al. introduced location- and time-specific carbon intensity metrics (Dodge et al., 2022; Sanvitto et al., 2023). Case studies such as BLOOM incorporated embodied emissions from hardware manufacturing (Luccioni et al., 2023), while open reports like Llama-2 (Meta AI Research, 2023) and OLMo (Groeneveld et al., 2024) disclosed approximate training footprints, providing transparency for reproducible energy studies. In parallel, a range of tools emerged to improve accounting. The ML $CO_2$ Impact Calculator required manual input (Lacoste et al., 2019), CodeCarbon extended this by embedding real-time monitoring into training workflows (Courty et al., 2024), CarbonTracker predicted emissions from early profiling (Anthony et al., 2020), Eco2AI integrated monitoring with PyTorch/TF (Kaack et al., 2022), and TracarB covered cluster-level usage (Valeye, 2021). While these tools increased transparency, they remain limited by narrow system boundaries, incomplete hardware coverage, and reliance on average rather than spatiotemporal grid factors.

Recent work has examined downstream deployment, including inference costs (Samsi et al., 2023; Luccioni et al., 2024), fine-tuning trade-offs (Wang et al., 2023), and system-level accounting frameworks such as CarbonConnect (Percy et al., 2024). Other studies evaluated optimisation strategies (Fernandez et al., 2025a), lifecycle impacts (Morrison et al., 2025), and called for stronger disclosure and policy integration (Luccioni et al., 2025; Fernandez et al., 2025b). While these efforts advanced discussions on efficiency, transparency, and governance, they largely address single models or isolated lifecycle stages. The broader ecosystem-level impact remains underexplored. In this paper, we move beyond case studies to systematically estimate the training emissions of thousands of models on Hugging Face, providing an industry-scale perspective on AI's carbon footprint and a baseline for tracking its future trajectory.

**Carbon accounting** refers to the systematic quantification and reporting of greenhouse gas (GHG) emissions, providing reliable foundations for climate policy and sustainability research. The Intergovernmental Panel on Climate Change (IPCC) established a comprehensive methodological framework in the 2006 Guidelines for National Greenhouse Gas Inventories, which has since been adopted by countries for sectoral inventories covering energy, industry, and agriculture (Eggleston et al., 2006). Within this framework, carbon accounting can be differentiated into industry-level accounting, which estimates total emissions from entire sectors throughout production, operation, and supply chains (IPCC, 2022; United States Environmental Protection Agency, 2023), and product-level accounting, which applies life-cycle assessment (LCA) to a single product or service across its full cradle-to-grave stages (Wor, 2011; ISO, 2018; myclimate, 2023; Tog, 2022).

Despite mature practices in other domains, few standardized frameworks exists for carbon accounting of the AI sector. The Software Carbon Intensity (SCI) (Green Software Foundation, 2024) published by the Green Software Foundation (GSF) defines a methodology for carbon accounting of a software system. It only measures the carbon intensity of a software application per functional unit, without using architecture-specific FLOPs or training metadata. Neither IPCC guidelines nor LCA standards extend to AI training or inference, and disclosure is largely absent. Recent steps, such as the EU AI Act, the Energy Efficiency Directive, California's AB 222, and ongoing ISO/IEC drafts (eua, 2024; EU2, 2023; AB2, 2025; ISO, 2025) – signal progress, but AI remains outside existing carbon accounting regimes.

**Emissions from AI training.** Recent studies have estimated the electricity use and carbon emissions of training large models, but typically focus on a few representative cases, leaving ecosystem-level impacts unclear. They have examined training emissions but treated FLOPs as a fixed computational quantity, rather than as part of the core indicator for evaluating carbon efficiency. Strubell et al.(Strubell et al., 2020) calculate training emissions using measured/reproduced electricity × regional EF for several NLP models (GPT-2, BERT,etc). Patterson et al.(Patterson et al., 2021) estimate FLOPs for Google models (T5, Meena,etc), but emissions are still derived from measured electricity × regional EF, not FLOPs-based estimation. Anthony et al.(Anthony et al., 2020) and Lacoste et al.(Lacoste et al., 2019) use FLOPs as a proxy for electricity consumption, without analyzing emissions-per-FLOP or cross-model carbon intensity. They consider hardware efficiency (FLOP/s), but none treat FLOPs as part of the standardized or comparable metric (e.g., Emission/FLOP) for carbon efficiency of AI models. Luccioni et al.(Luccioni et al., 2023) compute BLOOM's emissions from internal energy logs and regional EF. LLMCarbon(Faiz et al., 2023) infers energy use

during training from flops, detailed hardware and parallelism configurations, and validates its model on a small set of fully-specified LLMs. However, prior work either focuses on single-model or single-architecture case studies(Strubell et al., 2020; Luccioni et al., 2023; Wang et al., 2023; Morrison et al., 2025), depends on complete metadata or internal telemetry (Patterson et al., 2021), or provides experiment-level monitoring tools (Lacoste et al., 2019). They face challenges to scale thousands of models and enable reproducible, platform-wide carbon attribution. The key bottleneck, overlooked in past works, lies in estimating FLOPs, hardware, region, PUE, and runtime for thousands of heterogeneous models with missing disclosures.

Complementary tools exist: Hugging Face introduced a `co2_eq_emissions` field in 2022 (covering only ∼0.12% of repositories). This field relies on CodeCarbon (Courty et al., 2024), which requires detailed runtime logging of hardware power and grid intensity. CodeTracker(Anthony et al., 2020) similarly monitors real-time CPU/GPU power draw during model training and estimates the resulting carbon emissions based on the local grid intensity. It requires full runtime access, hardware telemetry, and controlled training environments, and therefore cannot be applied to large open-source ecosystems such as Hugging Face. Consequently, CodeCarbon and CodeTracker both remain limited for large-scale assessments without complete training metadata.

Taken together, these efforts underscore that AI training generates substantial emissions, but existing evidence remains fragmented and insufficient for understanding the aggregate impact. Snapshots of isolated models or voluntary disclosures cannot capture the scale of emissions produced across tens of thousands of models now hosted and shared globally. Without broader and more systematic estimates, it is difficult to assess the true magnitude of AI's carbon footprint or to design effective mitigation strategies. To address this gap, we turn to Hugging Face, the largest open repository of AI models, as a vantage point for constructing model-level training emission estimates at scale.

## 3 ESTIMATING CARBON EMISSIONS OF HUGGING FACE MODELS

Hugging Face hosts more than two million models, of which approximately 1.7 million are publicly accessible. Many entries are re-uploads, format conversions, or quantized variants that do not involve new training, while others lack essential training information. After filtering, we retained widely used models, resulting in 5,234 models with more than 5,000 downloads. Our primary analysis focuses on this >5,000 group.

### 3.1 IDEAL RUNTIME-BASED ESTIMATION MODEL

In an ideal scenario, if the computational power of the supercomputer used for training is known ($P_{\text{comp}}$), together with the total training time ($T_{\text{comp}}$) and the carbon intensity of electricity in the training region ($EF_{\text{region}}$, measured in $\text{kgCO}_2/\text{kWh}$), the training-related emissions can be estimated as

$$E_{\text{train}} = P_{\text{comp}} \times T_{\text{comp}} \times EF_{\text{region}}. \tag{1}$$

However, very few models disclose such information, and accurate data on the carbon footprint of supercomputing centers is even harder to obtain. Therefore, alternative strategies are required.

**Estimating Computational Power.** We approximate the effective computational power of the supercomputer through the following decomposition:

$$P_{\text{comp}} \approx N_{\text{GPU}} \times P_{\text{GPU}}^{\text{eff}} \times \text{PUE}, \tag{2}$$

where $N_{\text{GPU}}$ denotes the number of GPUs employed during training, and $P_{\text{GPU}}^{\text{eff}}$ represents the effective average power draw per GPU (in kW). We define $P_{\text{GPU}}^{\text{eff}} = P_{\text{GPU}} \times R_{\text{eff}}$, where $P_{\text{GPU}}$ is the nominal or rated power consumption of the GPU (often approximated by its Thermal Design Power, TDP), and $R_{\text{eff}}$ is a runtime utilization factor that accounts for the gap between theoretical peak and actual workload efficiency. The term PUE stands for the Power Usage Effectiveness of the data center, which accounts for the additional overhead of cooling and infrastructure and typically ranges between 1.2 and 1.7 (CAE Lighting, 2025).

**Estimating Training Time.**  The training time is estimated based on the overall computational workload required, expressed in floating-point operations (FLOPs). For a given model, the total training FLOPs is denoted by $F_{\text{train}}^{\text{total}}$. Assuming knowledge of GPU throughput, the base training time can be approximated as

$$T_{\text{base}} = \frac{F_{\text{train}}^{\text{total}}}{\theta_{\text{GPU}} \times N_{\text{GPU}} \times R_{\text{eff}}}, \tag{3}$$

where $\theta_{\text{GPU}}$ is the sustained throughput per GPU in FLOPs per second (e.g., $3.12 \times 10^{14}$ FLOPs/s for NVIDIA A100 SXM under TF32), $N_{\text{GPU}}$ is the number of GPUs, $R_{\text{eff}}$ is the runtime utilization efficiency. Since training often involves restarts, debugging, and warm-up cycles, we incorporate a time amplification factor $A_{\text{time}} \geq 1$, yielding $T_{\text{comp}} = T_{\text{base}} \times A_{\text{time}}$.

**Final Estimation Model.**  Combining Eq. 1 2 and 3, the training-related carbon emissions of Hugging Face models can be estimated as

$$E_{\text{train}} \approx \underbrace{\left(N_{\text{GPU}} \times P_{\text{GPU}} \times R_{\text{eff}} \times \text{PUE}\right)}_{P_{\text{comp}}} \times \underbrace{\left(\frac{F_{\text{train}}^{\text{total}}}{\theta_{\text{GPU}} \times N_{\text{GPU}} \times R_{\text{eff}}} \times A_{\text{time}}\right)}_{T_{\text{base}}} \times EF_{\text{region}} \tag{4}$$

$$= \frac{P_{\text{GPU}}}{\theta_{\text{GPU}}} \times \text{PUE} \times F_{\text{train}}^{\text{total}} \times A_{\text{time}} \times EF_{\text{region}}.$$

Eq. 4 represents our estimation framework for model-level training of carbon emissions on Hugging Face. It is physically consistent and captures the key drivers of training-related emissions: $\frac{P_{\text{GPU}}}{\theta_{\text{GPU}}}$ is effective energy per FLOP. $F_{\text{train}}^{\text{total}}$ reflects model size and training iterations. PUE represents data center overhead, accounting for cooling and distribution losses. $A_{\text{time}}$ as time amplification factor captures parallelization inefficiencies, communication overhead, and system-level delays. $EF_{\text{region}}$ translates consumed energy into carbon emissions based on the local electricity mix. In short, Eq. 4 decomposes training emissions into *hardware × efficiency × computation × system amplification × infrastructure × environment*.

**AI Training Carbon Intensity.**  While the direct estimation of training emissions is informative, it may not always be intuitive for practitioners. Eq. 4 provides a simplified framework for quantifying training emissions, and it can be further abstracted by grouping all factors except $F_{\text{train}}^{\text{total}}$ into a single coefficient. We define this coefficient as the *AI Training Carbon Intensity (ATCI)*, which represents the average carbon emission per FLOP of computation:

$$\text{ATCI} \approx \frac{P_{\text{GPU}}}{\theta_{\text{GPU}}} \times \text{PUE} \times A_{\text{time}} \times EF_{\text{region}}. \tag{5}$$

Similar to the regional emission factor $EF_{\text{region}}$, which translates electricity use into carbon emissions based on grid composition, ATCI translates FLOPs into carbon emissions by integrating hardware efficiency, data center overhead, runtime amplification, and regional carbon intensity. In other words, ATCI can be interpreted as the effective "carbon cost per compute" for AI training.

In our work, we estimate ATCI across a large collection of HF models and report empirical values. This offers the community a practical reference point, enabling researchers and practitioners to approximate the training-related carbon footprint of models even in the absence of complete system-level disclosures. To further validate and calibrate this index, we regress observed training emissions on FLOPs, emission factors, and hardware families (Figure 2, Appendix A.8.2). This regression provides empirical evidence that supports the ATCI formulation: the estimated FLOPs elasticity of $\sim$0.8 confirms a near-linear scaling of emissions with compute, while significant hardware-specific effects highlight the role of accelerator efficiency in shaping ATCI. ATCI serves as a theoretical abstraction of carbon cost per compute, and the regression results act as an empirical cross-check.

**Carbon intensity of regional grids among models.**  To estimate the carbon emissions associated with model training and deployment, we assign each model a regional electricity carbon intensity

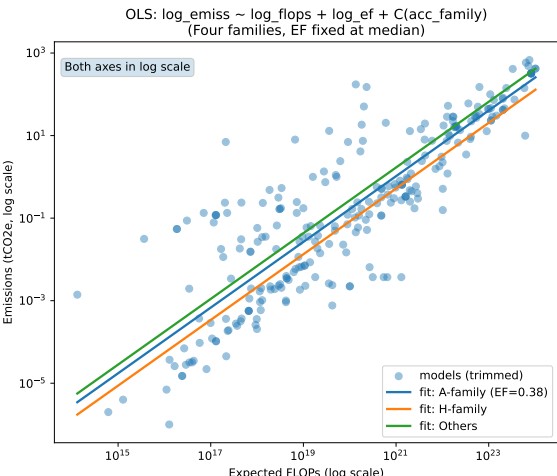

Figure 2: Scatter plot of estimated training emissions versus expected FLOPs, with regression fits for different accelerator families (A: NVIDIA A100/A800, H: H100/H800, Others). Both axes are log-scaled. The fitted model is $\log(E_{\text{train}}) = -39.25 + 0.85\log(\text{EF}_{\text{region}}) + 0.83\log(F_{\text{train}}^{\text{total}}) - 0.83\,I\{\text{H-family}\} + 0.63\,I\{\text{Others}\}$. Results indicate $\sim 0.83$ FLOPs elasticity. Relative to the A-family (baseline), the H-family shows about $56\%$ lower emissions. The "Others" exhibits roughly $88\%$ higher emissions. A unified PUE and time amplification factor are assumed due to missing data center disclosures.

based on the best available geographic information: (i) When the model card specifies the training region or provides a specific emission factor, that value is used directly. (ii) In the absence of such disclosures, the region is inferred from the training organization's compute infrastructure or institutional affiliations. Regional factors follow the *Carbon Intensity of Electricity Generation* dataset from Our World in Data (Ritchie et al., 2025). In cases where region information is also missing or indeterminate, we use the global average carbon intensity of $0.445$ tCO$_2$/MWh, consistent with IEA guidelines (IEA, 2024).

## 3.2 TRAINING FLOPS ESTIMATION

A central quantity in our framework is the total training compute $F_{\text{train}}^{\text{total}}$, expressed in FLOPs. For transformer-based NLP models (e.g., BERT, GPT, LLaMA), we use the standard FLOPs approximation, FLOPs $\approx c \times N_{\text{params}} \times N_{\text{tokens}}$, where $c$ reflects the relative cost of attention and feedforward operations. Empirical studies suggest $c$ typically falls in the range 5–8, and we adopt $c = 6$ as a conservative baseline, while sensitivity analyses with an extended range (5–12) are reported in Appendix A.3 and A.6. For computer vision (CV) and multimodal models, we apply architecture-specific heuristics. For Vision Transformers (ViTs) and CLIP models, FLOPs are estimated from patch embeddings and Transformer blocks, with training FLOPs approximated as six times the single-step inference cost; for CLIP, we apply a $1.1\times$ adjustment to account for the language branch. For diffusion models (e.g., Stable Diffusion, DiT), FLOPs are calculated by summing the convolution, self-attention, and cross-attention costs across denoising steps. For large multimodal Transformers that process image-text tokens with LLM-like backbones, we approximate compute as FLOPs $\approx 6 \times N_{\text{params}} \times N_{\text{tokens}}$, analogous to NLP models. Details of architecture-specific formulas, corrections for fine-tuning and Mixture-of-Experts structures, and our imputation strategy for missing parameters are provided in the Supplement (Appendix A.6–A.7).

## 3.3 HANDLING MISSING VALUES

**Three-tier Strategies.** Emission estimation relies on partially disclosed information, which we cross-validate against multiple sources. We adopt a three–tier framework: 1) Tier 1 with rich disclosures (hardware type, GPU hours, or FLOPs). Emissions are computed from electricity use (GPU hours $\times$ power $\times$ grid factor) and from FLOPs–based inference, serving as calibration points (Appendix A.8.1); 2) Tier 2 with partial disclosures (e.g., FLOPs only). We impute missing values using representative hardware efficiencies and average overheads (Appendix A.8.2). Representative cases in Figure 2 also show how disclosure profiles map to estimation strategies and how regressions link FLOPs to emissions across hardware generations; 3) Tier 3 with minimal information (e.g., parameters only). Emissions are approximated via parameter–based regressions (Appendix A.8.3).

## 3.4 UNCERTAINTY PROPAGATION

Our estimation framework involves several quantities that carry measurement or imputation uncertainty. Since these variables enter multiplicatively in Eqs. (3)–(6), we propagate uncertainty using the standard first-order relative-error formulation for products:

$$\frac{\Delta E}{E} \approx \sqrt{\sum_i \left(\frac{\Delta x_i}{x_i}\right)^2}, \tag{7}$$

where $x_i \in \{F_{\text{train}}^{\text{total}}, P_{\text{GPU}}, \theta_{\text{GPU}}, A_{\text{time}}, \text{PUE}, EF_{\text{region}}\}$. The expression in Eq. (7) shows that the uncertainty in $E_{\text{train}}$ is governed by the combined relative errors of the multiplicative factors that define the training emissions. The resulting uncertainty structure is summarized in Appendix A.2.

# 4 RESULTS

## 4.1 TRAINING EMISSION RESULTS

Reporting results follow standard significant-digit rules: aggregate emissions are given with at most two significant digits. Thus, our estimates indicate that, as of August 2025, training 5,234 models with more than 5,000 downloads has resulted in cumulative emissions of approximately $5.8 \times 10^4$ tCO$_2$e with an uncertainty of $\pm 2 \times 10^4$ tCO$_2$e, consistent with the propagated error in Eq. 7 (See details in Appendix A.2). We compare average ATCI and model-level emissions across modalities and training types in Table.1.

**CV & multimodal exhibit higher training emission intensity than NLP.** CV's average ATCI is 0.16 tCO$_2$e/EFLOP versus NLP's 0.14 tCO$_2$e/EFLOP, indicating that per unit compute of vision training tends to translate into more energy and carbon. This gap plausibly comes from heavier data pipelines and lower hardware efficiency in vision workloads (e.g., large image/video batches, augmentation, diffusion/decoder-only VAEs, and higher I/O/memory pressure that reduces accelerator utilization), as well as the prevalence of multi-stage training (pretrain + alignment + SFT) for VLMs.

**Emission differences between foundation models (or individual models) and finetuned models.** The results highlight a clear divergence between Foundation & Individual models and Finetuned models in both emission intensity and their aggregate climate footprint. Finetuned models exhibit a higher mean ATCI (0.22 vs. 0.14 t/EFLOPs), suggesting that each unit of computation in downstream training typically incurs greater carbon emissions. This pattern aligns with the typical deployment environments: large foundation and standalone models are often trained on centralized, energy-efficient clusters with optimized hardware utilization and cleaner grid mixes, whereas fine-tuning workloads are more widely distributed across smaller-scale, less efficient, and often metadata-poor computing environments, which inflates per-EFLOP carbon intensity. Despite their higher ATCI, finetuned models contribute only a minor share of the total emissions, as the computational scale of foundation-model pre-training overwhelmingly dominates. Overall, while finetuning tends to be "dirtier per EFLOP," the majority of AI's training-related carbon footprint is still driven by a relatively small number of extremely compute-intensive foundation-model runs.

Table 1: Emission indicators and repository counts.

(a) Model-level CO$_2$e Emission Indicators

| Category | Mean ATCI (t/EFLOPs) | Mean (t) | Total ($10^4$ t) |
|---|---|---|---|
| Foundation & Individual | 0.14 | 12 | 5.5 |
| Finetuned models | 0.22 | 8 | 0.3 |
| CV & Multi-Modal | 0.16 | 11 | 2.3 |
| NLP | 0.14 | 11 | 3.5 |

(b) Repository Counts by Tier (downloads> 5000)

| Tier | NLP Repos | CV/MM Repos |
|---|---|---|
| Tier 1 | 390 | 352 |
| Tier 2 | 944 | 1679 |
| Tier 3 | 3053 | 220 |

**ATCI.** We further interpret the significance of ATCI, defined as the ratio of training emissions to floating-point operations. ATCI captures the carbon efficiency of model training pipelines, abstracting away from model size or absolute compute cost, and therefore provides a normalized metric

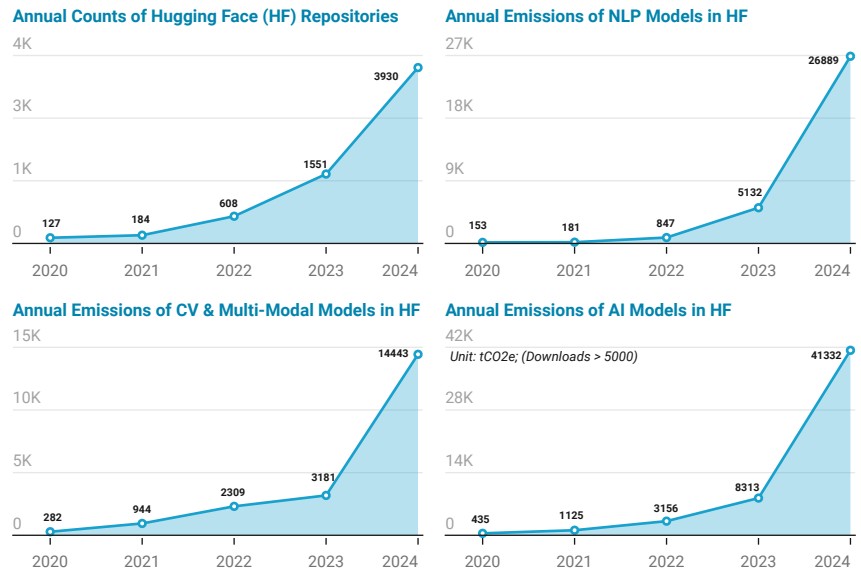

Figure 3: Annual training emissions of AI models (downloads 5,000+) in HF from 2020 to 2024

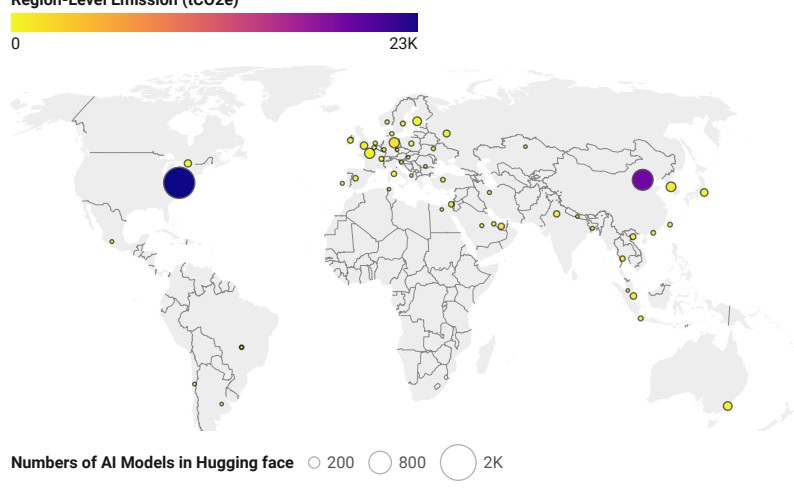

Figure 4: Global accumulative training emissions of AI models (downloads 5,000+).

to compare across modalities and training paradigms. Overall, the results highlight that (a) modality matters: vision/multimodal training is more carbon-intensive per compute; (b) lifecycle practice matters: finetuned variants exhibit higher per-checkpoint emissions not only because they undergo repeated downstream training and alignment cycles, but also because they are typically run on less energy-efficient hardware environments, whereas a small number of large foundation-model pretrains still dominate the aggregate carbon footprint. Model-level ATCI provides a meaningful measure for understanding the estimated environmental burden of AI training, as well as the relative efficiency differences among model classes.

## 4.2 HUGGING FACE TRAINING EMISSION ACROSS REGION AND TIME

**Region.** As shown in Figure 4, regional aggregation reveals an uneven distribution of training emissions. The United States dominates the landscape ($2.3 \times 10^4$ tCO$_2$e across 1,000+ repositories), followed by China ($1.9 \times 10^4$ tCO$_2$e; 404 repositories). In contrast, most European countries (e.g., the United Kingdom, France, Italy, Finland), as well as Canada and Australia, host many repositories but generate comparatively small emissions per model, indicating lighter-weight workloads or lower-compute research practices.

**Temporal evolution.** As shown in Figure 3, training emissions of models (downloads 5000+) on Hugging Face have escalated sharply over time. From 2020–2021 to 2024–2025, annual emissions increased from only $\sim 4.3 \times 10^2$ t$CO_2$e to more than $4.1 \times 10^4$ t$CO_2$e, reflecting nearly two orders of magnitude growth within five years. The composition of these emissions also shifted substantially. Early periods were dominated by CV and multi-modal models, but NLP activity expanded rapidly between 2022 and 2024, becoming the largest contributor during this interval. In the most recent period (2024–2025), CV and multi-modal models once again surpassed NLP due to a surge in large-scale vision and multimodal releases. Together, these trends reveal both the accelerating pace of model training and the evolving distribution of computational demand across AI domains.

**Projected emission.** According to the projected electricity growth rate of global AI data centers (IEA, 2024), expected to rise from about $1.3\%$ of global electricity demand in 2024 to nearly $2.8\%$ by 2030 (and stabilising nearly $3.1\%$ by 2035). Figure 5 similarly illustrates a projected increase in model emissions. Our estimates show that models with over 5,000 downloads will grow from $\sim 4.1 \times 10^4$ in 2024 to $\sim 9.9 \times 10^4$ t$CO_2$e in 2035, those with over 1000 downloads from $\sim 5.8 \times 10^4$ to $\sim 1.4 \times 10^5$ t$CO_2$e, and the broader set with over 100 downloads from $\sim 1.0 \times 10^5$ to $\sim 2.5 \times 10^5$ t$CO_2$e.

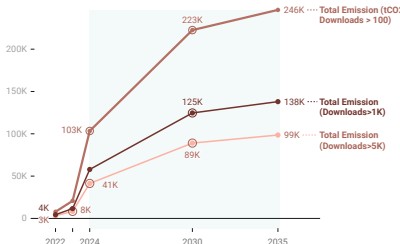

Figure 5: Projected training emissions of HF models at scale from 2024 to 2035.

### 4.3 CARBON DISCLOSURE QUALITY OF AI MODELS.

Among the more than two million repositories on Hugging Face, only 2,422 include a structured `co2_eq_emissions` field, and fewer than 200 provide any additional energy or emissions details in their `README`. In total, well under 0.2% of models disclose any environmental footprint, underscoring a substantial transparency gap (see Appendix A.9). However, even within the disclosed set, many entries suffer from inconsistent multi-source reporting and erroneous values, limiting their reliability. Table 2 highlights several representative model cases. It compares disclosed values from technical reports or Hugging Face metadata with our estimates, showing that our results are generally consistent with disclosures. Still, due to the lack of detailed disclosures for most models, we approximate missing quantities using industry or region-level averages, which inevitably introduces uncertainty. Nevertheless, cross-validation against the subset of disclosed models indicates that these estimation errors remain within an acceptable range (see Appendix A.3 and A.4). This underscores the feasibility of our approach and the urgent need for systematic, standardized reporting of emissions across the AI ecosystem.

Table 2: Illustrative comparison between disclosed and estimated training emissions.

| Model series | | | |
|---|---|---|---|
| Model series | Our Estimation | Disclosed emissions (t$CO_2$e) | Source |
| Llama 2 | 412 | 384 | Touvron et al. (2023) |
| CodeLlama | 72 | 65 | HF disclosed |

| Single model | | | |
|---|---|---|---|
| Model | Our Estimation | Disclosed emissions (t$CO_2$e) | Source |
| Meta Llama 2 (7B) | 33 | 31 | Touvron et al. (2023) |
| Meta Llama 2 (13B) | 52 | 62 | Touvron et al. (2023) |
| Meta Llama 2 (70B) | 327 | 291 | Touvron et al. (2023) |
| Meta-Llama 3 (70B) | 1,010 | 1,900 | HF disclosed |
| Meta Llama 3.1 405B | 8,176 | 8,930 | AI Index (2025) |
| Bloom | 24.7 | 24.7 | Luccioni et al. (2023) |
| OLMoE-1B-7B-0924 | 20 | 18 | Morrison et al. (2025) |
| stable-diffusion-v1 | 13.3 | 11.25 | HF disclosed |
| sam-vit-base | 2.7 | 2.80 | HF disclosed |
| sam2-hiera-small | 4.67 | 3.89 | HF disclosed |
| bioclip | 0.20 | 0.13 | HF disclosed |
| stable-diffusion-2 | 17 | 15 | HF disclosed |
| stable-video-diffusion-img2vid | 13 | 19 | HF disclosed |
| stable-diffusion-v1-5 | 13.50 | 11.25 | HF disclosed |

**Error on Models With Disclosed Emissions**  To evaluate the accuracy of our framework against ground-truth disclosures, we analyze 292 models that publicly report their total training emissions (see Appendix.A.3). To ensure robustness, we exclude unreliable disclosures and numerically unstable cases, and adopt a robust trimming procedure to mitigate the impact of heavy-tailed outliers. Relative errors are defined as $\text{RE}_i = |\hat{E}_i - E_i|/E_i$. To obtain a stable evaluation less affected by extreme outliers, we perform symmetric trimming,

Table 3: Robust evaluation on models with disclosed emissions.

| Metric | Value |
|---|---|
| MAPE | 0.42 |
| Median RE | 0.32 |
| Hit rate ($\times 2$ / $\times 3$) | 0.74 / 0.82 |

retaining the central 95% of samples by excluding the lowest and highest 5% of relative-error values. The evaluation yields the results in Table.3. The results indicate that, despite a few extreme outliers, the majority of models exhibit stable and accurate emission estimates, with approximately 74% and 82% of models falling within $\times 2$ and $\times 3$ of their disclosed values, respectively.

## 5 CONCLUSIONS

This paper presents a FLOPs-based framework to estimate training-related carbon emissions of Hugging Face models at scale. Our analysis shows that even within the open-source ecosystem, cumulative training emissions already reach the order of $10^4$–$10^5$ tons of $CO_2$e, comparable to the footprint of a medium-sized country over several weeks. This highlights both the urgency of standardized disclosure and the value of open repositories as anchors for industry-scale carbon accounting.

**Limitation and Future Work.**  Our study presents the systematic accounting of training-related carbon emissions for mainstream models hosted on Hugging Face. These results provide a useful reference point for researchers, practitioners, and the public in understanding the environmental costs of AI. At the same time, several important limitations remain, highlighting directions for future work. First, our analysis focuses exclusively on open-source models. A large fraction of the most influential models are proprietary, and their training processes and energy consumption remain undisclosed. Existing reports suggest that these closed-source models may contribute substantially to overall emissions, likely exceeding the footprint of the open-source community. Second, we focus only on training emissions. Yet training is only one part of the picture. Research activities that do not yield a final deployed model also consume considerable resources, and inference at deployment scale is expected to dominate AI's long-term energy demand. Understanding the emissions from inference workloads will require complementary approaches, such as analyzing data center expansion, hardware deployment statistics, and the size of the inference services market. Third, our study does not attempt to capture the full lifecycle emissions of AI systems. A complete assessment would account for the embodied carbon from hardware manufacturing, research and experimentation, model training, and deployment-scale inference, as well as the accounting and attribution of such emissions across stakeholders. Developing standardized methodologies for lifecycle carbon accounting in AI remains an open and urgent challenge.

**Extension to inference emission estimations.**  While our main analysis focuses on training, the framework can be extended to inference. The inputs can switch to inference-specific quantities: the power and throughput of the inference hardware (often different from training GPUs), the efficiency and batching characteristics of inference workloads, and the compute required per generated token. Once collecting these inputs, our framework can yield inference-emission estimates and inference emission intensity in exactly the same way as for training.

Overall, our work should be viewed as an initial step toward scalable estimation of training emissions. By quantifying the training emissions of a large body of open-source models, we provide an empirical anchor that future studies can extend toward closed-source models, inference workloads, and full lifecycle assessments. Such progress is essential for aligning AI development with sustainability goals and for informing the policy frameworks that will govern AI in the years ahead.

## REPRODUCIBILITY STATEMENT

We emphasize reproducibility as a key principle of this work. All reported results are based on open-source datasets that we collected and curated. To ensure transparency, we provide detailed descriptions of data collection, data cleaning, calculation, and estimation procedures in the Ap-

pendix.A.5, A.6, A.7, and A.8. The methods and evaluation protocols are described in the main text, and we will release both the datasets and the complete source code on GitHub upon publication to further facilitate verification and future research.

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

# A APPENDIX

## A.1 THE USE OF LARGE LANGUAGE MODELS (LLMS)

In this work, Large Language Models (LLMs) were used in two limited ways. First, we designed an LLM-based agent to assist with filtering and analyzing parts of the Hugging Face README data, which supported the pre-processing of model metadata. Second, LLMs were used for light editing and polishing of the manuscript text to improve clarity and readability. No core research ideas, experimental design, or final analysis depended on LLM output.

## A.2 SOURCES OF ESTIMATION ERROR ACROSS MODELS

Our framework assigns training-emission estimates to three disclosure levels (Tier 1–3), each of which introduces uncertainty from different sources. This section details the origin and nature of these uncertainties, and how they propagate into the final emission estimates.

**Tier-1: Fully or Partially Disclosed Training Metadata** Tier-1 models provide the most reliable information and fall into two subcategories.

**(a) Direct disclosure.** Some models report one or more of electricity consumption (MWh) or $CO_2e$ emissions; GPU/TPU-hours; explicit accelerator type and count; training region or datacenter provider. In these cases, emissions follow the standard power–time formulation

$$E_{\text{T1}} \approx \text{MWh} \times EF_{\text{region}}, \tag{8}$$

with uncertainty dominated only by reporting granularity (rounding, coarse region labels).

**(b) High-confidence FLOP-based Tier-1.** For other Tier-1 models, total training FLOPs are disclosed or recoverable with high fidelity (e.g., from official technical reports), and emissions are computed as

$$E_{\text{T1}} \approx F_{\text{train}}^{\text{total}} \times K_{\text{eff}} \times EF_{\text{region}}. \tag{9}$$

Here, $K_{\text{eff}}$ represents the effective electricity consumption per unit of compute:

$$K_{\text{eff}} = \frac{P_{\text{GPU}} \times A_{\text{time}}}{\theta_{\text{GPU}} \times \text{peakTFLOPS}}, \tag{10}$$

where $P_{\text{GPU}}$ is the average power draw, $\theta_{\text{GPU}}$ the achieved utilization efficiency, and $A_{\text{time}}$ a runtime amplification factor capturing communication, I/O, and other overheads. Uncertainty therefore propagates primarily through small variations in $\theta_{\text{GPU}}$, $A_{\text{time}}$, and regional emission factors. Because both $F_{\text{train}}^{\text{total}}$ and the hardware family are well constrained, Tier-1 FLOP-based estimates also exhibit low uncertainty.

**Tier-2: FLOPs Known, Hardware and Runtime Partially Missing** Tier-2 models disclose (or allow reconstruction of) the total training FLOPs, but lack full hardware/runtime information. Emissions are therefore computed as

$$E_{\text{T2}} \approx F_{\text{train}}^{\text{total}} K_{\text{eff}} EF_{\text{region}}, \tag{11}$$

where $K_{\text{eff}}$ groups accelerator throughput, datacenter amplification, PUE, and average power.

Tier-2 uncertainty thus arises from:

1. Imputed hardware family (A100/A800/H100/TPU/AMD),
2. Throughput/efficiency variance in $\theta_{\text{GPU}}$ across implementations and parallelism setups,
3. Datacenter amplification uncertainty ($A_{\text{time}}$),
4. Regional EF uncertainty due to missing or ambiguous geography.

Because FLOPs is known while $K_{\text{eff}}$ and $EF_{\text{region}}$ are imputed, Tier-2 inherits moderate uncertainty.

**Tier-3: Neither FLOPs Nor Runtime Disclosed** Tier-3 models require the heaviest imputation. Total FLOPs must be estimated from model parameters via a scaling-law style approximation:

$$F_{\text{train}}^{\text{total}} \approx c \, N_{\text{params}}, \tag{12}$$

where the coefficient $c$ implicitly absorbs typical choices of token counts, training stages (pretraining, SFT, RLHF), number of epochs, and curriculum details for a given family of models.

Emissions then follow:

$$E_{\text{T3}} \approx (c \, N_{\text{params}}) K_{\text{eff}} EF_{\text{region}}. \tag{13}$$

Major sources of Tier-3 uncertainty include:

1. Scaling-law coefficient variance (the proportionality constant $c$ is architecture- and corpus-specific and absorbs variation in effective token counts and training stages);
2. Hardware inference as in Tier-2 (accelerator family, utilization, and datacenter amplification folded into $K_{\text{eff}}$);
3. Regional EF uncertainty when geography is missing or coarse;
4. Compounded multiplicative propagation across $F_{\text{train}}^{\text{total}}$, $K_{\text{eff}}$, and $EF_{\text{region}}$.

Since both $F_{\text{train}}^{\text{total}}$ and $K_{\text{eff}}$ must be imputed, and each term enters multiplicatively, Tier-3 accumulates the largest theoretical error. Plugging representative relative uncertainties as shown in Table.4 into Eq. 7 yields

$$\frac{\Delta E}{E} \approx \sqrt{\left(\frac{\Delta F}{F}\right)^2 + \left(\frac{\Delta K_{\text{eff}}}{K_{\text{eff}}}\right)^2 + \left(\frac{\Delta EF}{EF}\right)^2} \sim 0.9\text{–}1.5, \tag{14}$$

corresponding to an implied Tier-3 uncertainty range of $\pm(90\text{–}150)\%$, i.e., roughly **2–3×** variation for typical models.

Table 4: Typical relative-uncertainty ranges for multiplicative factors in Eqs. (3)–(6).

| Quantity | Symbol | Typical Relative Error ($\Delta x/x$) |
|---|---|---|
| Total training FLOPs (Tier-1/2) | $\Delta F/F$ | 0.05–0.15 |
| Total training FLOPs (Tier-3 proxy $cN_{\text{params}}$) | $\Delta F/F$ | 0.60–0.80 |
| GPU average power draw | $\Delta P/P$ | 0.05–0.10 |
| Utilization efficiency | $\Delta\theta/\theta$ | 0.10–0.25 |
| Runtime amplification factor | $\Delta A_{\text{time}}/A_{\text{time}}$ | 0.10–0.20 |
| Regional emission factor | $\Delta EF/EF$ | 0.10–0.20 |

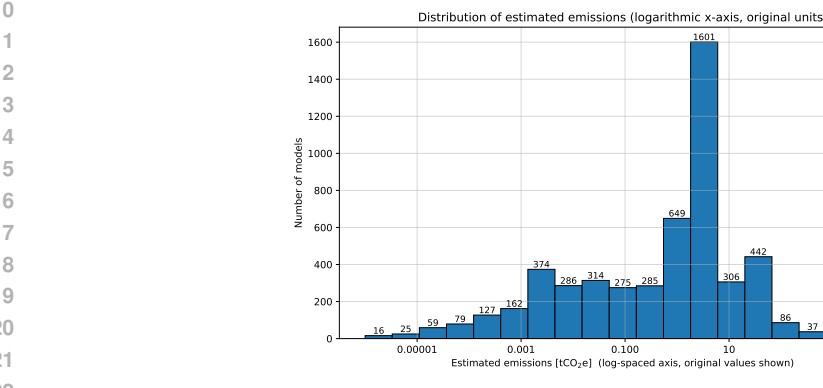

Figure 6: Distribution of Estimated Emissions of Hugging Face Models (5,000+ Downloads)

**Summary of Error Sources and Expected Magnitudes**

- **Tier-1 (low)**: $\pm 5$–$15\%$ (direct or high-confidence FLOPs-based; minimal imputation).
- **Tier-2 (moderate)**: $\pm 40$–$70\%$ (hardware, efficiency, and EF imputation; FLOPs accurate).
- **Tier-3 (high)**: $\pm 90$–$150\%$ (both FLOPs proxy $cN_{\text{params}}$ and hardware/datacenter effects imputed; multiplicative compounding).

These theoretical ranges follow directly from the multiplicative structure in Eqs. (3)–(6) , the first-order propagation rule (Eq. 7), and representative relative uncertainties as shown in Table.4. The theoretical ranges are also consistent with our pseudo-missingness experiments in Appendix A.4.

**Significant-digit rules.** Reporting results follow standard significant-digit rules: aggregate emissions are given with at most two significant digits, and uncertainty intervals with one significant digit. For ATCI, we apply the same significant–digit principles. Because ATCI is a ratio of two quantities with comparable relative uncertainty (emissions and FLOPs). Accordingly, ATCI values are reported with one to two significant digits, matching the precision justified by the input factors and the error structure in Eq. 7.

**Aggregate Uncertainty.** Considering the expected uncertainty of each tier (Tier 1: 10%, Tier 2: 55%, Tier 3: 120%) by their respective emissions proportions (Tier 1: 33%, Tier 2: 60%, Tier 3: 7%), we yield an aggregate-level uncertainty of approximately $\pm 40\%$. Thus, our estimates indicate that, as of August 2025, training 5,234 models with more than 5,000 downloads has resulted in cumulative emissions of approximately $5.8 \times 10^4$ $tCO_2e$ with an uncertainty of $\pm 2 \times 10^4$ $tCO_2e$

### A.3 UNCERTAINTY ANALYSIS OF TRAINING-EMISSIONS ESTIMATION

This section provides a comprehensive analysis of estimation uncertainty in the HUGGINGCARBON framework. We evaluate the uncertainty from four complementary perspectives: (i) metadata disclosure sparsity on Hugging Face, (ii) comparison against disclosed FLOPs, runtime, and emissions, (iii) relative error on models with disclosed emissions, and (iv) variance-based decomposition of uncertainty sources.

**Metadata Disclosure Landscape** We evaluate metadata disclosure across all 5,234 models in our dataset. To construct this metadata repository, we systematically collected training-related information from three classes of sources:

1. **Official Hugging Face model cards**, including structured fields (e.g., `compute_used`, `hardware`, `carbon_emissions`), author-provided notes, and embedded configuration snippets.
2. **Repository configuration files**, such as `config.json`, tokenizer/vision encoder configs, and architecture descriptors. These files provide parameter counts, layer depths, hidden sizes, patch sizes, and other FLOPs-relevant attributes.

3. **External authoritative sources**, including official technical reports, GitHub repositories, and arXiv papers referenced in the model cards. When multiple sources were available, we applied a deterministic priority order (direct disclosure → config-derived → paper-derived → regression-estimated).

Table 5: Metadata disclosure sparsity across the Hugging Face models (5,000+ downloads).

| Metadata Field | Count | Disclosure Rate |
|---|---|---|
| Training emissions (tCO$_2$e) | 292 | 5.58% |
| Electricity use (MWh) | 15 | 0.29% |
| Grid emission factor | 5 | 0.10% |
| Training region | 54 | 1.03% |
| GPU type | 955 | 18.25% |
| TPU Pod | 159 | 3.04% |
| Training runtime hours | 179 | 3.42% |
| Training device count | 414 | 7.91% |

Only 5–6% of models disclose energy or emission-related metadata. This structural sparsity is the primary source of uncertainty in open-source carbon accounting. For all models with any disclosed training information (FLOPs, electricity use, grid factors, or total emissions), we have compiled a detailed comparison table containing disclosed quantities and our reconstructed estimates.

**Summary of Models with Self-Disclosed Emissions**   The set of 292 models that self-disclosed their training emissions includes series such as *Bloom*, *CodeLlama*, *Stable Diffusion*, *SAM/SAM2*, and *BioCLIP*; recent Meta Llama 3/3.1/3.2 and Llama 4 variants (e.g., *meta-llama/Llama-3.1-405B*, *meta-llama/Llama-3.1-70B*, *Meta-Llama-3-70B*); AllenAI's *OLMo* and *OLMoE* models (e.g., *allenai/OLMo-7B-hf*, *allenai/OLMo-2-1124-13B-Instruct*); EleutherAI's *GPT-NeoX-20B*; image and video models from Stability AI (e.g., *stable-diffusion-2*, *stable-video-diffusion-img2vid* and related variants); a large cluster of biomedical language models from the *OpenMed* organization; and smaller models such as *ModernBERT* variants, rerankers, and tiny classifiers.

**Variance-Based Decomposition of Uncertainty Sources**   We model training emissions as:
$$E \approx \text{FLOPs} \times \text{EF} \times K,$$
where EF is the regional emission factor and $K$ absorbs hardware efficiency, runtime, and PUE effects. For each model, we infer $K_{\text{eff}} = E/(\text{FLOPs} \times \text{EF})$ and perform a variance-based sensitivity analysis with realistic perturbations:

- FLOPs: $\pm 30\%$ uncertainty,
- Hardware/runtime/PUE: $\pm 20\%$,
- Grid EF: $\pm 10\%$.

Using 1,000 Monte Carlo samples per factor, we estimate each source's contribution to $\text{Var}(E)$. The global contributions averaged across all models are:

Table 6: Variance-based uncertainty decomposition.

| Uncertainty Source | Variance Share |
|---|---|
| FLOPs estimation | 66% |
| Hardware/runtime/PUE ($K$) | 27% |
| Grid emission factor (EF) | 7% |

FLOPs estimation constitutes the dominant uncertainty driver, while EF accounts for only a small fraction. Based on variance decomposition across estimation components, FLOPs estimation contributes ∼66% of overall uncertainty, hardware assumptions ∼27%, and grid emission factors ∼7%. These results demonstrate that uncertainty arises primarily from ecosystem-wide metadata sparsity rather than methodological limitations.

## A.4 Pseudo-Missingness Experiment for Tier 2 and Tier 3 Uncertainty

To explicitly quantify the uncertainty introduced by Tier 2 and Tier 3 estimation, we conduct a **pseudo-missingness experiment** that closely aligns with real metadata disclosure patterns observed on Hugging Face.

**Ground-truth selection.** We use **all Tier 1 models** as high-confidence ground truth, including those with direct energy disclosure or those with complete metadata (training hardware and training GPU hours). To avoid numerical instability in relative errors, we remove only trivial-emission cases, eliminating numerical artifacts while preserving essentially all meaningful Tier 1 models.

**Constructing pseudo Tier 2 / Tier 3 samples.** We randomly sample 70% of Tier 1 models and **artificially mask metadata** to simulate realistic missingness:

- **Pseudo Tier 2:** retain FLOPs, emission factor, and GPU family; mask hardware type, runtime, and direct/disclosed energy.
- **Pseudo Tier 3:** further remove FLOPs, leaving only parameter count, emission factor, and GPU family.

These masked models are re-evaluated using the **exact Tier 2 and Tier 3 regression pipelines** described in the paper. Predicted emissions are compared with Tier 1 ground truth using absolute error (AE) and relative error (RE). Results are shown in Table 7.

Table 7: Pseudo-missingness experiment results for Tier 2 and Tier 3 uncertainty.

| Pseudo Tier | n | MAE ($tCO_2e$) | Median RE | P90 RE |
|---|---|---|---|---|
| Tier 2 (FLOPs-based) | 312 | 61.62 | **0.57** | **1.20** |
| Tier 3 (Params-based) | 123 | 111.42 | **0.99** | **1.92** |

- **Tier 2 estimates remain highly stable:** median RE $\approx 0.57$; 90% of predictions exhibiting $\sim 1.2\times$ relative error.
- **Tier 3 remains informative despite minimal metadata:** median RE $\approx 0.99$; 90% within $\sim 2\times$ relative error.

Median RE summarizes the **typical multiplicative deviation** introduced when metadata is partially or severely missing. For example, a Median RE of $0.57$ indicates that half of the reconstructed emissions differ from the Tier 1 ground truth by no more than $57\%$, while the remaining half may exhibit larger deviations.

In this context, Median RE captures how much accuracy can be preserved when Tier 1-quality metadata is downsampled to the more realistic, incomplete metadata available under Tier 2 or Tier 3 conditions. A low Median RE for pseudo Tier 2 suggests that FLOPs and emission factors alone are sufficient to retain a substantial fraction of estimation fidelity. These results show that Tier 2 and Tier 3 estimates are not exact but remain **predictive at the order-of-magnitude level under realistic missingness patterns**.

**Additional mitigation mechanisms.** To constrain uncertainty, our framework incorporates:

- architecture-based FLOPs derivation and runtime backsolving with bounded parameter ranges,
- GPU-family regression calibrated on Tier 1 ground-truth models,
- variant deduplication to avoid double-counting mirrors or lightweight derivatives,

Together, these mechanisms ensure that Tier 2 and Tier 3 predictions remain anchored to validated Tier 1 models and provide stable, interpretable estimates across the open-source model ecosystem.

## A.5 Data Collection and Processing Pipeline

**Automated Crawling.** We collect heterogeneous metadata from Hugging Face model repositories and associated documentation. The crawler reads repository descriptors (`README.md`, model cards, metadata CSVs, configs.json) and extracts candidate fields including *hardware type*, *GPU/TPU counts*, *training duration*, and especially *training FLOPs*. For FLOPs disclosures, we implemented robust parsing functions that can handle varied numeric expressions (e.g., shorthand "2k", "1.2M", or scientific notation such as "$5 \times 10^{21}$"), ensuring standardized floating-point values for downstream estimation. All extracted fields are normalized and stored in structured CSV/JSON tables, providing a consistent basis for regression analysis and emission estimation.

**Repository Deduplication.** To avoid double-counting emissions from mirrored repositories, we applied a systematic deduplication rule: when both an official repository and an `unsloth/...` mirror exist, the mirror is dropped unless the discrepancy in reported values is negligible ($\leq 0.1\%$), in which case the `unsloth` version is retained as canonical. In addition, we excluded derivative artifacts such as GGUF or quantized models (e.g., 4bit/8bit, AWQ, GPTQ/PTQ/NF4/FP8/Q4/Q5) since they represent deployment optimizations rather than independent training runs. These filters ensure that only unique, training model entries are preserved in the dataset.

**Agent Workflow.** To handle inconsistent disclosures and missing fields, we developed an LLM-based agent workflow (GPT-4o) that performs: (i) **hardware recognition**, mapping noisy or aliased strings to canonical GPU/TPU families; (ii) **unit normalization**, distinguishing between wall-clock hours and GPU-hours using contextual cues; (iii) **cross-file integration**, employing a dedicated **web search agent** to locate and retrieve corresponding technical reports or project website released by model developers, which were then cross-validated against Hugging Face metadata and incorporated into the final dataset. We merge all findings with regional emission factor datasets. Ambiguous cases (e.g., extreme FLOPs values, unclear unit conventions) were flagged for manual inspection by human annotators.

**Human Verification.** To ensure reliability, five independent human annotators reviewed a stratified subsample of repositories. They checked accelerator mappings, parsed FLOPs statements, and validated whether durations corresponded to GPU-hours or wall-clock hours. Annotators resolved edge cases such as conflicting information across README text and metadata tables. Inter-annotator agreement was calculated to calibrate the agent's confidence thresholds.

**Data Integration.** All sources (GPU/TPU metadata, FLOPs estimates, and regional emission factors) were merged into unified tables via normalized identifiers. Duplicate columns and conflicting values were harmonized, and each record carries diagnostic notes (e.g., method of estimation, source of FLOPs, reasons for imputation). This enables transparent traceability of every emission estimate. The final dataset consists of harmonized records with accelerator type, count, training duration (direct or imputed), FLOPs used, power draw, regional EF, and estimated emissions ($tCO_2e$). All records include provenance notes indicating whether values were obtained via direct disclosure, agent inference, or human annotation.

## A.6 NLP Training FLOPs Estimation: pretraining vs. finetuning with optimization-aware corrections

OpenAI's scaling law study (Kaplan et al., 2020) introduced the widely used approximation for training compute of large-scale language models:

$$\text{FLOPs} \approx c \times N_{\text{params}} \times N_{\text{tokens}},$$

where $N_{\text{params}}$ is the number of model parameters, $N_{\text{tokens}}$ the number of training tokens, and $c$ a constant reflecting the balance between attention and feed-forward operations. Empirical evidence suggests $c$ typically falls in the range 5–8, depending on architecture and training configuration.

In our framework, we refine this baseline approximation to account for model heterogeneity and practical training regimes:

- **Architecture type.** Encoder-only models (e.g., BERT), decoder-only models (e.g., GPT, LLaMA), and encoder–decoder models (e.g., T5, BART) differ in the ratio of feed-forward to attention compute, which shifts $c$ within the baseline range of 5–8.

- **Parameter-efficient fine-tuning (PEFT).** For methods such as adapters and LoRA, only a fraction of parameters are trainable. We therefore rescale the effective parameter count to reflect $N_{\text{trainable}}$, while partially accounting for frozen weights that still incur forward-pass compute during backpropagation.

- **Mixture-of-Experts (MoE).** For MoE architectures, dense parameter count does not represent the actual compute cost. We instead replace $N_{\text{params}}$ with the number of *active* parameters per token, determined by the top-$k$ experts selected during routing, and introduce a routing overhead correction.

To encompass these variations, we extend the coefficient range to 5–12 based on recent empirical studies, ensuring coverage of both standard transformer training and specialized regimes such as PEFT and MoE. Unless otherwise specified, we adopt $c = 6$ as a conservative baseline for the main analysis, while sensitivity analyses over the full range are reported in this supplement.

In practice, we estimate training compute (FLOPs) for transformer-based NLP models by combining structural information with training configuration metadata extracted from Hugging Face model cards, repository documentation, and associated papers. This process is automated in our analysis pipeline and implemented in several steps:

**1) Model classification and parameter extraction.** Each model is classified as encoder-only (e.g., BERT), decoder-only (e.g., GPT, LLaMA), or encoder–decoder (e.g., T5). When available, we directly record the number of trainable parameters ($N_{\text{params}}$). If parameters are missing, we infer them from architecture descriptors such as hidden size, number of layers, and attention heads.

**2) Effective parameter count adjustments.** For pretraining we set $N_{\text{params}}$ to the full parameter count. For others, we distinguish:

- **Full-parameter Fine Tuning (FT)**: $N_{\text{params}}$ is the full count.

- **Parameter-efficient FT (PEFT)** (e.g., LoRA/adapters): we substitute $N_{\text{params}}$ by the number of *active trainable* parameters $N_{\text{trainable}}$ and include a forward-pass reuse factor since frozen weights still incur inference-side compute during backprop. Concretely,

$$\text{FLOPs}_{\text{base,PEFT}} \approx c_{\text{arch}}\big(\alpha_{\text{frozen}}\, N_{\text{frozen}} + N_{\text{trainable}}\big) \times N_{\text{tokens}},$$

with $\alpha_{\text{frozen}} \in [0.2, 0.5]$ reflecting the proportion of frozen-path compute amortized in backward (empirical, task- and stack-dependent).

- **Mixture-of-Experts models**: we substitute the full parameter count with the number of active parameters per token, i.e., the sum of dense parameters and the top-$k$ experts activated per forward pass. Here, we replace $N_{\text{params}}$ by the *active* parameters per token, i.e.,

$$N_{\text{params}}^{\text{MoE}} \approx N_{\text{dense}} + \underbrace{k \cdot \frac{N_{\text{experts}}}{E}\, N_{\text{expert}}}_{\text{top-}k \text{ experts per token}},$$

where $k$ is the top-$k$ routing, $E$ is the number of experts per layer, and $N_{\text{expert}}$ the per-expert parameters. We also apply a routing overhead factor $\alpha_{\text{route}} \in [1.00, 1.05]$ and optional load-imbalance penalty if reported (Lepikhin et al., 2020; Fedus et al., 2022).

**3) Token accounting.** When $N_{\text{tokens}}$ is not directly reported, we infer it from dataset size and epochs, or reconstruct it from step geometry:

$$N_{\text{tokens}} \approx S \times G, \tag{1}$$

$$\text{where} \quad G = W \times A \times L \times B. \tag{2}$$

Here $S$ denotes the total number of training steps, $W$ the world size (number of devices), $A$ the gradient accumulation steps, $L$ the average sequence length, and $B$ the per-device batch size.

**4) Baseline FLOPs estimate.**  Let $N_{\text{params}}$ denote the number of (active) trainable parameters and $N_{\text{tokens}}$ the number of training tokens effectively processed. The baseline lower-bound follows (Kaplan et al., 2020):

$$\text{FLOPs}_{\text{base}} \approx c_{\text{arch}} \times N_{\text{params}} \times N_{\text{tokens}},$$

where $c_{\text{arch}} \in [5, 12]$ accounts for architectural differences in the ratio of attention and feed-forward compute. In our implementation we set

$$c_{\text{arch}} = \begin{cases} c_{\text{enc}} & \text{encoder-only,} \\ c_{\text{dec}} & \text{decoder-only,} \\ c_{\text{encdec}} & \text{encoder–decoder,} \end{cases} \quad \text{with } c_{\text{enc}}, c_{\text{dec}}, c_{\text{encdec}} \in [5, 12].$$

We use $c_{\text{arch}} \in [5, 12]$: encoder-only and decoder-only models default to 6, while encoder–decoder models use 7, with flexibility for further adjustments.

**5) Optimization- and system-aware corrections.**  We multiply the baseline by factors capturing optimizer, precision, memory-saving, and parallelism overheads/efficiencies:

$$\text{FLOPs} = \text{FLOPs}_{\text{base}} \times \alpha_{\text{opt}} \, \alpha_{\text{prec}} \, \alpha_{\text{ckpt}} \, \alpha_{\text{act}} \, \alpha_{\text{pipe}} \, \alpha_{\text{dp}} \, \alpha_{\text{misc}}.$$

Default ranges (when explicit telemetry is absent) are:

- **Optimizer** $\alpha_{\text{opt}}$: Adam/AdamW maintain moments (extra pointwise ops), typically 1.10–1.20; Adafactor closer to 1.05; SGD 1.00.

- **Numerical precision** $\alpha_{\text{prec}}$: bf16/fp16 kernels often match theoretical FLOPs ($\approx 1.00$); fp32 $\approx 1.10$ due to bandwidth/latency effects; fp8 with scale management 0.90–1.00 (model- and kernel-dependent).

- **Activation checkpointing** $\alpha_{\text{ckpt}}$: recomputation overhead 1.05–1.30 (depth/segment length dependent).

- **Activation sparsity / fused kernels** $\alpha_{\text{act}}$: fused-attention, FlashAttention, bias-drop, etc. can yield 0.90–0.98 effective factor (stack-sensitive).

- **Parallelism** $\alpha_{\text{pipe}}, \alpha_{\text{dp}}$: pipeline bubbles and data-parallel sync yield 1.00–1.10 each in typical steady state.

- **Misc. serving/training stack** $\alpha_{\text{misc}}$: graph capture/JIT (benefit) vs. logging, mixed dataloading (overhead), default 0.98–1.05.

These factors encode the empirical observation that theoretical compute systematically underestimates realized costs due to software and hardware under-utilization (Fernandez et al., 2025a).

## A.7 MULTIMODAL TRAINING FLOPs ESTIMATION

For multimodal models, we employ an architecture-specific methodology to estimate training FLOPs. Our automated analysis pipeline categorizes models into several primary architectures, including Vision Transformers (ViT), Contrastive Language-Image Pre-Training (CLIP) models, Convolutional Neural Networks (CNNs), Diffusion models, and Transformers. The core of this approach is extracting key architectural parameters from HuggingFace model cards and configuration files. For CNNs, however, we directly run the model with a randomized input tensor of a unified resolution to precisely calculate the single-step inference FLOPs.

Notably, this analysis excludes the computational cost of parameter-efficient fine-tuning (PEFT) techniques, such as LoRA and other adapters. While increasingly prevalent for model customization, the compute required for training these modules is typically several orders of magnitude smaller than that of full model pre-training or fine-tuning, rendering its contribution negligible in our large-scale carbon footprint assessment.

With $E$ as the number of training epochs and $I$ as the number of training images per epoch, we apply the following tailored estimation strategies for different architectures:

**1) ViT and CLIP models.** For Vision Transformer (ViT) based models, we first calculate the FLOPs for a single forward step by summing the contributions from the patch embedding layer and the subsequent Transformer blocks. Let $H, W, P, C$ be the input image height, width, patch size, and channels, respectively, and let $d, L, r$ be the model's hidden dimension, number of layers, and MLP expansion ratio. The number of input tokens is $N = \frac{H \cdot W}{P^2} + 1$ (including the [CLS] token).

The total MACs (Multiply-Accumulate operations) for one forward pass can be broken down as:

- **Patch Embedding**: $M_{embed} = H \cdot W \cdot C \cdot d$
- **Transformer Block**: The computation is dominated by the multi-head self-attention (MHSA) and the MLP layers, where $M_{MHSA} = 4Nd^2 + 2N^2d$ and $M_{MLP} = 2rNd^2$.

Thus, the total MACs for one single step of a ViT model can be expressed as:

$$M_{ViT} = M_{embed} + L \cdot (M_{MHSA} + M_{MLP}) = HWCd + L[(4 + 2r)Nd^2 + 2N^2d] \quad (3)$$

Based on the common heuristic that training FLOPs are approximately six times the inference MACs (accounting for a $3\times$ factor for the training procedure and a $2\times$ factor for converting MACs to FLOPs), the final FLOPs are:

$$F_{ViT} = 6 \times E \cdot I \cdot M_{ViT} \quad (4)$$

For CLIP models, we approximate the computational cost of the language branch as $10\%$ of the vision branch. Therefore, we apply a $1.1\times$ factor to the ViT result:

$$F_{CLIP} = 1.1 \times F_{ViT} \quad (5)$$

**2) Diffusion models.** For U-Net-based models (e.g., Stable Diffusion), the MACs for a single denoising step are calculated by summing the compute across all layers in the U-Net's down-sampling, middle, and up-sampling blocks. This includes contributions from 2D convolutions ($M_{conv}$), self-attention ($M_{SA}$), and cross-attention ($M_{CA}$) layers. The total FLOPs are then estimated as:

$$F_{Diffusion} = 6 \times E \cdot I \cdot (M_{conv} + M_{SA} + MCA) \quad (6)$$

For Diffusion Transformer (DiT) models, the calculation is analogous to that of ViT. The total FLOPs for a single step can be estimated by the sum of the patch embedding, the stack of $L$ Transformer blocks. The core computation within each DiT block, which includes self-attention, optional cross-attention, and an MLP, follows the same principles as the ViT block calculation.

$$F_{DiT} = 6 \times E \cdot I \cdot M_{DiT} = 6 \times E \cdot I \cdot [M_{embed} + L \cdot (M_{MHSA} + M_{MLP})] \quad (7)$$

**3) Transformers.** For Transformer-based models such as large vision-language models, where the architecture is predominantly a large language model processing multimodal tokens, the total training FLOPs are approximated as:

$$F_{Transformers} = 6 \times N \cdot D \quad (8)$$

where N represents the number of model parameters and D is the total number of tokens in the training data.

**Data Imputation Strategy.** Our automated pipeline may encounter models with incomplete configurations that lack the parameters necessary for FLOPs estimation. In such cases, we implement a prototype-based imputation strategy. Specifically, we pre-select a canonical or widely-recognized "prototype model" for each major architectural category (e.g., google/vit-base-patch16-224-in21k for ViTs). When a model is found to have missing parameters, the pipeline populates the missing fields with the corresponding values from the prototype model. For models where FLOPs cannot

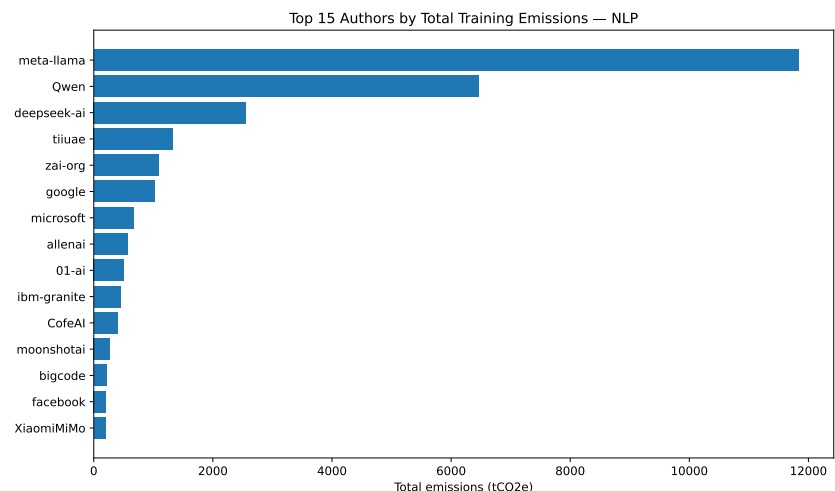

Figure 7: Top 15 Authors with Highest Estimated Training Emissions of Hugging Face NLP Models (5,000+ Downloads)

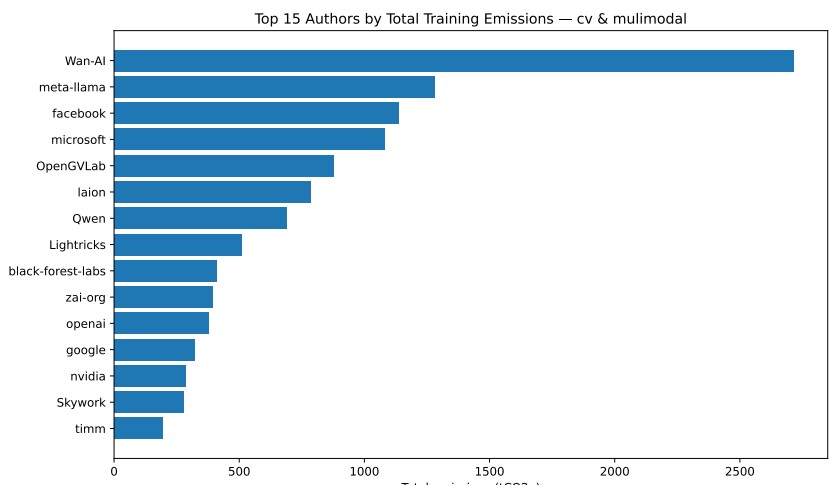

Figure 8: Top 15 Authors with Highest Estimated Training Emissions of Hugging Face CV & Multi-Modal Models (5,000+ Downloads)

be calculated at all (e.g., due to a missing configuration file), we impute the final FLOPs value using the mean of all other models in the same category. This approach ensures the robustness and comprehensive coverage of our estimation process.

### A.8 EMISSION ESTIMATION: CONSIDERING MISSING VALUE

To accommodate heterogeneous levels of disclosure across model repositories, we adopt a three–tier framework for training emission estimation:

- **Tier 1: Rich disclosures.** Models provide sufficient information required in Appendix. A.6 and Appendix.A.6 that is either directly disclosed or can be directly computed, such as *hardware type* (GPU/TPU family), reported *training GPU hours*, and/or total training FLOPs. In these cases, training duration and energy use can be established with the highest accuracy, enabling reliable emission estimation.
- **Tier 2: Partial disclosures.** Models have reported information for estimating the total FLOPs used in training, without hardware details or runtime information. Here, we estimate training emissions by assuming representative hardware efficiency values and average

system overhead factors, mapping FLOPs into energy consumption under a standardized configuration.(see Appendix A.8.2)

- **Tier 3: Minimal disclosures.** Models report only the parameter count, with no FLOPs or hardware details available. For these cases, we rely on a parameter–based regression (see Appendix A.8.3) as a fallback, using cross–sectional elasticity estimates to approximate emissions from model scale.

Table 8: Three–tier framework for handling missing values in training emission estimation.

| Tier | Available Information | Estimation Method & Accuracy |
|------|----------------------|------------------------------|
| 1 | Hardware type, GPU hours, or total FLOPs (606 models) | Direct electricity use (GPU hours × power × grid factor) and FLOPs–based inference; *High* (calibration set) |
| 2 | FLOPs available but no hardware/runtime details | FLOPs mapped to energy using representative hardware efficiency and overheads; *Medium* |
| 3 | Parameter counts only | Parameter–based regression to approximate FLOPs and emissions; *Low* |

### A.8.1 EMISSION ESTIMATION PIPELINE WITH TIER 1 MODELS

We implement a unified estimator that integrates accelerator recognition, multi-node topology, overhead factors, and regional emission intensities to approximate training-related carbon emissions. The pipeline is designed to handle heterogeneous disclosures across model repositories, including cases with incomplete or ambiguous hardware information.

**Hardware Normalization and Accelerator Imputation Procedure**   In the implementation, accelerators are mapped to a small set of *canonical families* with associated peak TFLOPS, average power, and efficiency: NVIDIA A100 / A100-80GB / A100-64GB / A800, H100 / H200 / H800, V100, A40, A30, T4, L4, RTX 6000 ADA, AMD MI250X / MI300X, and Google TPU V2 / V3 / V4 / V5E / V5P. Assignment proceeds as follows.

For Tier 1 (disclosed hardware), when model cards report training hardware type, these strings are used directly. If traininghardwaretype indicates TPU, the pod name (e.g., "v4-128", "v3-8") is parsed and mapped to a canonical TPU family; if the generation cannot be resolved, TPU V3 is used as a mid-range default. Otherwise, the device is treated as a GPU and is normalized using regex rules, matching patterns; the matched family is then used to look up peak TFLOPS, average power, and efficiency.

For Tier 2 and Tier 3 (imputed hardware), when metadata is incomplete, models with TPU hardware but ambiguous pod strings are assigned TPU V3 as a conservative default, and models known to use GPUs but lacking a resolvable training gpu type fall back to an A100-class assumption (A100 peak TFLOPS, ∼0.30–0.35 efficiency, 400 W power) as a representative datacenter GPU.

AMD and TPU jobs are therefore not collapsed into NVIDIA families: MI250X and MI300X have their own TFLOPS/power entries, and TPUs are handled via dedicated TPU families. Only when no reliable family can be inferred do we use an A100-class default for GPUs or TPU V3 for TPUs, keeping assumptions conservative and internally consistent. Peak compute throughput (TFLOPs/s) and average power consumption are tabulated for major GPU and TPU families under FP16/BF16 tensor-core settings. Custom mappings standardize diverse naming conventions (e.g., "A100 80GB", "TPUv4-8"), while TPU pod descriptors are canonicalized into TPU V2/V3/V4/V5E/V5P. Throughput efficiency is set by accelerator type as shown in Table.9.

**System Overheads.**   We include cluster-level overheads beyond accelerator power: (i) an IT overhead factor (20% relative to GPU draw) covering CPU/RAM/NIC usage, (ii) fixed per-node power (250 W), and (iii) per-node network overhead (100 W). A unified PUE of 1.2 accounts for datacenter infrastructure inefficiency.

Table 9: Canonical Accelerator Families Used in Estimation

| Accelerator Family | Peak TFLOPs | Avg. Power (W) | Efficiency |
|---|---|---|---|
| A100 | $3.12 \times 10^{14}$ | 400 | 0.35 |
| A100 80GB | $3.12 \times 10^{14}$ | 400 | 0.35 |
| A100 64GB | $3.12 \times 10^{14}$ | 400 | 0.35 |
| A800 | $3.12 \times 10^{14}$ | 350 | 0.30 |
| H100 | $9.89 \times 10^{14}$ | 600 | 0.45 |
| H200 | $1.00 \times 10^{15}$ | 650 | 0.45 |
| H800 | $8.00 \times 10^{14}$ | 550 | 0.40 |
| V100 | $1.25 \times 10^{14}$ | 300 | 0.25 |
| T4 | $6.5 \times 10^{13}$ | 70 | 0.20 |
| L4 | $1.20 \times 10^{14}$ | 75 | 0.25 |
| A40 | $3.00 \times 10^{14}$ | 300 | 0.25 |
| A30 | $1.65 \times 10^{14}$ | 300 | 0.25 |
| RTX 6000 ADA | $1.45 \times 10^{14}$ | 300 | 0.25 |
| MI250X | $3.83 \times 10^{14}$ | 560 | 0.30 |
| MI300X | $1.20 \times 10^{15}$ | 750 | 0.40 |
| TPU V2 | $4.5 \times 10^{13}$ | 120 | 0.25 |
| TPU V3 | $1.23 \times 10^{14}$ | 187 | 0.35 |
| TPU V4 | $2.75 \times 10^{14}$ | 220 | 0.45 |
| TPU V5E | $8.0 \times 10^{13}$ | 120 | 0.35 |
| TPU V5P | $2.90 \times 10^{14}$ | 280 | 0.45 |

**Input Integration.** The estimator merges three data sources: (a) GPU/TPU metadata (type, count, nodes, duration), (b) expected FLOPs from scaling estimates or disclosures, and (c) regional emission factors (tCO$_2$/MWh).

**Runtime Attribution.** Two pathways are implemented:

1. **Direct runtime:** If *training hours* are disclosed, emissions are computed directly from reported wall-clock or GPU-hours multiplied by hardware power draw.

2. **Imputed runtime:** If training duration is *not* disclosed but total FLOPs are available, we back-compute runtime as

$$T = \frac{F_{\text{train}}}{\text{PeakTFLOPs} \times R_{\text{eff}} \times N_{\text{acc}}},$$

where $F_{\text{train}}$ is expected FLOPs, $R_{\text{eff}}$ is throughput efficiency, and $N_{\text{acc}}$ is accelerator count. This ensures models with only FLOPs disclosure can still be assigned a plausible runtime estimate.

If neither hours nor FLOPs are available, the case is labeled `insufficient`, which is then categorized as a tier 2 or tier 3 model.

**Emission Calculation.** Total energy consumption is given by

$$MWh = \Big(P_{\text{acc}} \cdot N_{\text{acc}} \cdot T + \text{IT overhead} + \text{node/network fixed}\Big) \times \text{PUE},$$

where $P_{\text{acc}}$ is average power per accelerator, $N_{\text{acc}}$ the accelerator count, and $T$ the effective training duration (hours). Multiplying by the regional emission factor yields emissions in tCO$_2$e.

A.8.2 EMISSION ESTIMATION WITH TRAINING FLOPS FOR TIER 2 MODELS.

We establish a log–log regression between model training FLOPs, regional emission factors, and hardware accelerator families:

$$\log(E_i) \ = \ \beta_0 \ + \ \beta_1 \log(F_i) \ + \ \beta_2 \log(\mathrm{EF}_i) \ + \ \sum_k \gamma_k \, \mathbf{1}\{\mathrm{acc}_i = k\} \ + \ \varepsilon_i,$$

where $E_i$ denotes the training emissions (tCO$_2$e), $F_i$ the expected FLOPs, $\mathrm{EF}_i$ the grid emission factor (tCO$_2$/MWh) in the model training region, and $\mathbf{1}\{\mathrm{acc}_i = k\}$ an indicator for accelerator family $k$. The regression yields a robust elasticity of $\beta_1 \approx 0.83$ for FLOPs, and $\beta_2 \approx 0.85$ for grid emission factors, while hardware differences are captured by the categorical terms $\gamma_k$.

Thus, the approximation logic can be expressed as

$$E_i \ \approx \ C \cdot F_i^{0.83} \cdot \mathrm{EF}_i^{0.85} \cdot \delta(\mathrm{acc}_i),$$

where $C = \exp(\beta_0)$ is a constant and $\delta(\mathrm{acc}_i)$ is a multiplicative adjustment depending on the accelerator family.

Table 10: OLS regression of log-emissions on FLOPs, grid emission factors, and hardware dummies. Robust (HC3) standard errors in parentheses.

| Variable | Coefficient | Std. Error |
|---|---|---|
| Intercept | $-39.252^{***}$ | (1.685) |
| $\log(F)$ | $0.829^{***}$ | (0.034) |
| $\log(\mathrm{EF})$ | $0.847^{**}$ | (0.362) |
| acc[T.H-family] | $-0.827^{**}$ | (0.331) |
| acc[T.Others] | $0.629$ | (0.389) |
| $^{***}p < 0.01, \,^{**}p < 0.05, \,^{*}p < 0.1$ | | |

### A.8.3 EMISSION ESTIMATION WITH PARAMETERS FOR TIER 3 MODELS.

This parameter-based regression is used as a fallback for Tier-3 models, where no additional information is available to support FLOPs-based estimation. We further establish a log–log regression between model parameter counts, regional emission factors, and model subtype categories:

$$\log(E_i) \ = \ \beta_0 \ + \ \beta_1 \log(P_i) \ + \ \beta_2 \log(\mathrm{EF}_i) \ + \ \gamma \, \mathbf{1}\{\mathrm{subtype}_i = \mathrm{finetune}\} \ + \ \varepsilon_i,$$

where $E_i$ denotes the training emissions (tCO$_2$e), $P_i$ the parameter count of the model, $\mathrm{EF}_i$ the grid emission factor (tCO$_2$/MWh), and $\mathbf{1}\{\mathrm{subtype}_i = \mathrm{instruct}\}$ an indicator for instruction-tuned models.

The regression indicates an elasticity of $\beta_1 \approx 1.45$ with respect to parameters, while the effect of grid emission factors is smaller and statistically insignificant. Instruction-tuned variants show systematically lower emissions compared to base models.

Thus, the approximation logic can be expressed as

$$E_i \ \approx \ C \cdot P_i^{1.45} \cdot \mathrm{EF}_i^{0.34} \cdot \delta(\mathrm{subtype}_i),$$

where $C = \exp(\beta_0)$ is a constant and $\delta(\mathrm{subtype}_i)$ is a multiplicative adjustment depending on whether the model is instruction-tuned.

Table 11: OLS regression of log-emissions on parameter counts, grid emission factors, and subtype dummies. Standard errors in parentheses.

| Variable | Coefficient | Std. Error |
|---|---|---|
| Intercept | $-32.127^{***}$ | (1.139) |
| $\log(P)$ | $1.451^{***}$ | (0.054) |
| $\log(\mathrm{EF})$ | $0.343$ | (0.250) |
| subtype[T.instruct] | $-1.001^{***}$ | (0.297) |
| $^{***}p < 0.01, \,^{**}p < 0.05, \,^{*}p < 0.1$ | | |

## A.9 Self-Disclosed Emission in Hugging Face

| | Counts |
|---|---|
| All HF Repositories | 2,099,013 |
| Carbon Emission Disclosed in HF Carbon Emission Modules | 2,422 |
| Carbon Emission Disclosed in Readmes | 126 |

Figure 9: **Total Amount of Models with Self-Disclosed Emission in Hugging Face.** Out of more than 2.1 million repositories, only 2,422 include a structured carbon emissions field and just 126 mention energy use or emissions in their README files, highlighting a disclosure rate below 0.2%.

