# OpenReview forum: "Hugging Carbon: Quantifying the Training Carbon Emissions of AI Models at Scale"
_ICLR.cc/2026/Conference — ICLR 2026 Conference Desk Rejected Submission_

### Official Review · Reviewer_PKst · 2025-10-27

**Soundness:** 3
**Presentation:** 4
**Contribution:** 3
**Rating:** 8
**Confidence:** 5

**Summary:**

This paper presents an interesting FLOPs-based framework to estimate training emissions of open-source AI models on the Hugging Face Hub. They convert compute to energy based on hardware efficiency characteristics and then to emissions using readily available carbon intensity data -- they call this the AI Training Carbon Intensity and compute it for the most popular 2,097 models on the Hugging Face Hub.

**Strengths:**

The methodology used by the authors is interesting and relevant:
The three-tier strategy for estimating missing values makes sense given the amount of missing data and the lack of transparency in the field.

They also report novel results, for instance:
- [CV & multimodal exhibit higher training emission intensity than NLP] -- this is consistent with previous work on different modalities
- [Instruction-tuned models show higher median model-level emissions than base models] -- this is coherent with the fact that instruction tuning often happens iteratively, compared to base model training
- [The geographical bias of model training] - which is concentrated in the Global North (coherent with Abdalla et al.'s results)

**Weaknesses:**

There are quite a few assumptions that are made by authors that are not necessarily supported by empirical evidence, namely:

- The fact that models were trained at the geographical location of their authors -- many/most authors will use cloud compute in regions other than their own

- That "ATCI provides a meaningful measure for understanding not only the absolute environmental burden of AI training" -- since it is based on multiple estimations, the fact that it is "absolute" seems far-fetched to me.

- That authors provide a "comprehensive carbon accounting in AI" - when in fact they only estimate the emissions of training, and no other steps of the AI lifecycle.

Overall, I would suggest for the authors to use less superlative claims and focus on their concrete contribution and its strengths and weaknesses.

**Questions:**

The following statements are unclear to me:

- "instruction-tuned models show higher median model-level emissions than base models" -- do you mean for each model (summing together all the instruction tuning done for that model)? Otherwise I find it strange that base emissions are lower.

- What are the "architecture-specific heuristics" you apply to computer vision (CV) and multimodal models? how do you define them?

- What about smolLM architectures and others that don't follow the scaling-law approximation?

---

> ### Author Response · Authors · 2025-11-21
> **Response to Reviewer PKst (1/2)**
>
> Dear Reviewer PKst,
>
> Thank you for your helpful feedback and insightful suggestions. Our detailed responses are provided below.
>
> > **W1: "...models were trained at the geographical location of their authors -- many/most authors will use cloud compute in regions other than their own..."**
>
> **R1:** To assign regional carbon intensity, we use a sequential procedure designed to reflect the best available information.
> - If a model card specifies the training region or reports a region-specific emission factor, we take that value directly.
> - If this information is missing, we infer the region from the training organization’s compute infrastructure or institutional affiliations. This step is supported by an agent-based web search process that inspects each model's public documentation and related organizational profiles.
> - The Hugging Face author location is used only when neither of these sources is available.
>
> We have clarified the complete procedure for assigning regional carbon intensity in the revised manuscript (Section 3.2).
>
> To assess the effect of potential regional misclassification, we also conducted a sensitivity analysis of regional emission factor as shown in Appendix A.2. Substituting plausible regions typically changes the estimated emissions by **no more than 7%**, suggesting that our results can be robust to reasonable variations in regional assumptions.
>
>
> > **W2: "ATCI provides the absolute environmental burden of AI training -- since it is based on multiple estimations, the fact that it is "absolute" seems far-fetched to me."**
>
> **R2:** The wording has been revised accordingly. We no longer describe ATCI as the ''absolute'' environmental burden. Instead, the manuscript now characterizes ATCI as an **estimated** measure of training-related emissions that supports **relative comparisons** across model classes. The updated terminology has been incorporated into the revised version.
>
> ATCI is derived from disclosed or imputed parameters and therefore carries bounded uncertainty, which we quantify in Appendix A.3 using the pseudo-missingness experiment. When Tier-1 models are masked and re-estimated, Tier-2 predictions remain tightly bounded (median RE ≈ 0.57; 90% within ~1.2×), while Tier-3 predictions retain meaningful signal (median RE ≈ 0.99; 90% within ~2×). These results confirm that ATCI is an estimate whose uncertainty is **controlled and empirically characterized** across Tiers.
>
>
> > **W3: "a step towards comprehensive carbon accounting in AI -- when in fact they only estimate the emissions of training, and no other steps of the AI lifecycle."**
>
> **R3:** This work focuses specifically on training-related emissions. We have revised the text to state that our contribution is a framework for model-level training carbon estimation, rather than a full lifecycle assessment. Although other lifecycle stages such as inference are outside the present scope, the same data-driven estimation framework can be extended to the inference phase once reliable and comparable usage statistics (for example, request volumes, hardware profiles) become available.
>
> > **W4: "I would suggest for the authors to use less superlative claims and focus on their concrete contribution and its strengths and weaknesses."**
>
> **R4:** Thank you. This is a constructive suggestion. We will revise claims that could be interpreted as superlative (e.g., "comprehensive," "absolute") and instead focus on the concrete contributions.

---

> > ### Author Response · Authors · 2025-11-21
> > **Response to Reviewer PKst (2/2)**
> >
> > > **Q1: "instruction-tuned models show higher median model-level emissions than base models -- do you mean for each model (summing together all the instruction tuning done for that model)? "**
> >
> > **R1:** Our analysis compares **per-model training emissions**, evaluating each released model on Hugging Face individually. In the earlier formulation, an instruction-tuned model inherently incorporated the full cost of pretraining and the subsequent SFT or RLHF stages.
> >
> > To avoid potential ambiguity in this interpretation, we have revised the analysis to instead compare **(i) foundation and single-model releases** and **(ii) finetuned models** as two distinct categories. This update separates models trained from scratch (or released as standalone pretrained checkpoints) from models that include only SFT or RLHF stages without pretraining. The revised manuscript now reports this updated comparison in Table 1.
> >
> >
> > > **Q2: "What are the "architecture-specific heuristics" you apply to computer vision (CV) and multimodal models? how do you define them?"**
> >
> > **R2:** The architecture-specific heuristics used for CV and multimodal models are defined based on architecture labels extracted from Hugging Face. These labels are derived from model READMEs, HF metadata fields (such as `pipeline_tags` and `tags`), and configuration files.
> >
> > - **(i) Modality identification.** Using this metadata, models are first grouped by modality. NLP models include terms such as "text-generation" or "sentence-transformers." Computer vision models include tasks such as "image-classification" or "segmentation." Multimodal models include tasks such as "text-to-image," "image-text-to-text," or "image-to-text."
> >
> > - **(ii) Architecture classification.** To determine the underlying architectural type (e.g., ViT, CNN, CLIP, diffusion, Transformer), we used an LLM agent (GPT-4o) with web-search capabilities to retrieve and analyze each model's official documentation. The agent was prompted to output only the fundamental architecture type after inspecting the Hugging Face model card, ensuring a consistent labeling procedure prior to FLOP estimation.
> >
> > This process yields **611 Transformer models**, **420 ViT models**, **369 CNN-based models**, **269 diffusion models**, **186 CLIP/vision–language models**, and **47 DiT models**. These architecture labels determine the **architecture-specific FLOP heuristics** applied in our analysis. For example, ViT models use ViT FLOP formulas, CNNs use convolution-based FLOP formulas, and diffusion or CLIP models use U-Net plus text-encoder FLOPs. Full details are provided in Appendix A.7.
> >
> > > **Q3: "What about smolLM architectures and others that don't follow the scaling-law approximation?"**
> >
> > **R3:** We appreciate the reviewer's question. Our estimation framework does not rely on scaling-law assumptions about the relationship between model size, data volume, and performance. Instead of extrapolating from scaling trends, FLOPs are estimated using standard training‐practice heuristics. Once the architecture, parameter count, and approximate training-token volume are known or recoverable, the associated compute can be estimated.
> >
> > Regarding smolLM and other architectures, their training computations still follow the same transformer-style computational structure for a given parameter size. Therefore, FLOPs estimation remains applicable in practice. We acknowledge that a small subset of such models may deviate slightly from typical compute patterns, but they constitute only a minor portion of the open-source ecosystem and operate at substantially smaller scales. These deviations are unlikely to affect our aggregate results across thousands of models. We will add a brief note in the revised manuscript to clarify this point.

---

> > ### Comment · Reviewer_PKst · 2025-11-25
> >
> > Thank you for your response! I think you addressed my main concerns and questions.

---

### Official Review · Reviewer_yezs · 2025-10-29

**Soundness:** 2
**Presentation:** 2
**Contribution:** 2
**Rating:** 4
**Confidence:** 3

**Summary:**

This paper proposes a method and reproducible framework to estimate the carbon emissions due to training for many machine learning models (over two thousand open-source models from Hugging Face).  This framework leverages an approximation of the FLOPS (floating point operations) used to train the model, and translates a given number of FLOPS to an estimate of carbon emissions based on the GPUs likely used during training, the carbon intensity of the electric grid in which the model was likely trained, and the size of the model.  The latter translation is done using a metric of AI Training Carbon Intensity (ATCI) that the authors propose as a general metric for estimating the carbon emissions of AI models.  The authors conducted this analysis and reported estimates for many open-source models on Hugging Face, present top-level results in the paper, and will publish a data set and code to reproduce their analysis.

**Strengths:**

A key strength of this study is the scale of the results, providing carbon emissions estimates for thousands of commonly used open source models on Hugging Face.

Results give some nice evidence that NVIDIA H-series accelerators are more carbon efficient than A-series.  Furthermore, there is value in the ATCI metric for comparing model training pipelines — as the authors point out, ATCI provides a normalized metric to compare across modalities and training paradigms.

This work provides strong reason to advocate for better carbon disclosure quality of all widely-adopted AI models, as better disclosure of simple factors such as training GPU type, GPU-hours, and regions would greatly improve these estimates.

**Weaknesses:**

The paper states that “there is a lack of a dedicated set of measurement practices” — this statement strikes me as tenuous.  There is an established notion of Software Carbon Intensity published by the Green Software Foundation whose definition is exactly the AI Training Carbon Intensity defined in this paper, where the “functional unit” is scaled to a FLOP [1]

Section 3.1’s statement that models are classified into base and instruct models seems to imply that the models considered in this paper are LLMs, but as far as I can tell this is not explicitly stated.  Furthermore, Section 3.3 suggests the opposite, namely that the models considered include LLMs but also include e.g., Vision Transformers.

Assumptions are made throughout the analysis that are reasonable in the sense that they may be necessary due to the lack of data, but they are quite coarse-grained and thus they simplify the analysis a lot.
For example, in applying the scaling law approximation for transformers, a ratio c is set to capture the ratio between attention and feed-forward operations.  The authors select a uniform value of 6 in most of the analysis (sensitivity analysis is conducted in the Appendix, but this also seems like a single value c is applied to all models). Similarly, when estimating GPU power, it is assumed that the GPU operates at maximum TDP — recent papers have shown that LLM training (in particular) actually induces large fluctuations in GPU power [2], making this a simplification as well.

In my opinion, the main weakness of this paper is that the estimation model seems to heavily rely on these assumptions that are applied more or less uniformly to all models, so the final analysis is effectively a sophisticated unit conversion between model characteristics (e.g., size) and a very coarse estimate of emissions from public data.

I am generally happy to see a serious treatment of this estimation problem, since this paper does provide the first (to my knowledge) estimate of per-model training emissions — however, at the current stage, I do not see this as a sufficient contribution over existing guidelines and public data for ICLR.  I am happy to discuss further if I have misunderstood some key contributions of the paper.

1. https://sci.greensoftware.foundation
2. https://arxiv.org/abs/2508.14318

**Questions:**

How does the scaling law approximation (line 259) capture training time?  If I have two identical models, where one is trained for a month and the other is trained for 2 months, I would expect the latter model to have a carbon footprint twice as big.  Is that captured in this approximation?  If not, how is it captured in your analysis when GPU-hours are not reported?

Your estimate states 240 tCO2/e for Llama 3 / 3.1, but the Llama 3 repo on Hugging Face [3] states 2,290 tCO2/e.  This discrepancy could be because the paper is only considering a single model out of the Llama family, but this should be clarified.

Table 2 lists the disclosed emissions for some model series, but doesn’t include the estimations?

The types of accelerator families considered are not explicitly defined anywhere that I can find in the paper — the caption of Figure 2 suggests that NVIDIA A100/A800 and NVIDIA H100/H800 GPUs are considered.  How exactly are jobs assigned to one of these families?  For the Tier 1 data it sounds like this data is provided, but for Tier 2 it must be imputed, and even for Tier 1 data, a choice must be made if the model was trained using a different family of accelerator (e.g., AMD, Google TPU).  How is this handled?

Minor comment — it may make sense to report CO2e/FLOP in a differently-scaled metric such as grams of CO2e/FLOP, since reporting in tons gives small values that are difficult to conceptualize (e.g., 10^{-22} on line 309).

How is the grid region for carbon intensity chosen for each model?  Is it purely based on the country of the model’s origin?  Even within a single country, there can be a massive difference in the carbon intensity of the grid(s) — the US is a good example of this (e.g., compare California with Missouri).

3. https://huggingface.co/meta-llama/Meta-Llama-3-70B

---

> ### Author Response · Authors · 2025-11-21
> **Response to Reviewer yezs (1/4)**
>
> Dear Reviewer yezs,
>
> Thank you for your helpful comments. Our responses are provided below.
>
> > **W1: "The paper states that “there is a lack of a dedicated set of measurement practices” — this statement strikes me as tenuous. There is an established notion of Software Carbon Intensity published by the Green Software Foundation whose definition is exactly the AI Training Carbon Intensity defined in this paper, where the “functional unit” is scaled to a FLOP [1]"**
>
> **R1:** After examining the Software Carbon Intensity (SCI) specification, we find that SCI and our proposed AI Training Carbon Intensity (ATCI) operate at fundamentally different levels of analysis.
>
> - **SCI** is an energy-based, system-level metric intended for **software applications** during operation, with the functional unit typically tied to user-facing workload (e.g., requests or interactions). While the SCI framework is, in principle, flexible in how the functional unit is defined, its current guidance is geared toward service-level operation rather than internal AI training runs. It does not provide model-specific mechanisms for relating training FLOPs, hardware throughput efficiency, utilization, multi-node overhead, and training-pipeline design to carbon emissions. As such, SCI is not formulated as a **model-level training** emission metric.
>
> - **ATCI**, by contrast, is a FLOPs-normalized, model-centric metric specifically developed for AI **training**. It incorporates hardware characteristics, datacenter PUE, multi-node overhead, and region-specific emission factors, enabling the “carbon cost per FLOP” of training a particular model to be quantified. A key component of our framework is the large-scale estimation of **training FLOPs**, which relies on architecture-specific heuristics and represents a nontrivial contribution in its own right.
>
> In addition, our contribution extends beyond defining ATCI and estimating FLOPs. We develop a **scalable estimation framework** capable of imputing ATCI and total training emissions for **thousands of models** with heterogeneous and incomplete metadata.
>
> Our intention for this statement was to convey that, in the current era of large-scale AI, there are still **no internationally standardized measurement practices** for assessing the carbon footprint of **AI model training or inference**. In contrast, sectors such as agriculture and manufacturing rely on well-established international frameworks—such as ISO 14040/14044 for life-cycle assessment, the GHG Protocol for emission accounting, and ISO 50001 for energy management which provide clear and widely adopted methodological guidance. The AI domain currently lacks any comparable **model-level international protocols**, leaving training and inference emissions without standardized measurement practices.
>
> Our work aims to help fill this gap by introducing a FLOP-aligned, model-centric approach to estimating training emissions at scale. To improve clarity, we have revised the text and explicitly incorporated the SCI specification into the literature review, citing the Green Software Foundation’s definition in the updated manuscript.
>
>
> > **W2: "Section 3.1's statement that models are classified into base and instruct models seems to imply that the models considered in this paper are LLMs.... Section 3.3 suggests the opposite, namely that the models considered include LLMs but also include e.g., Vision Transformers."**
>
> **R2:** We thank the reviewer for pointing this out. The phrasing in Section 3.1 was not intended to imply that our dataset consists only of LLMs. Our analysis spans all major Hugging Face modalities, such as LLMs and computer vision models. The base and instruct terminology applies only to the LLM subset and was used solely to illustrate the two-stage nature of LLM training.
>
> To avoid ambiguity, we have revised the analysis to compare **(i) foundation and single-model releases** and **(ii) finetuned models** as two distinct categories. This revision unifies the treatment across NLP, CV, and multimodal models and separates models trained from scratch (or released as standalone pretrained checkpoints) from models that include only SFT or RLHF stages without pretraining. The updated comparison is now reported in Table 1.
>
> We have also revised the text to explicitly clarify that our dataset includes both pretraining checkpoints and the large population of downstream finetuned models, which together are essential for understanding environmental impacts.

---

> > ### Author Response · Authors · 2025-11-21
> > **Response to Reviewer yezs (2/4)**
> >
> > > **W3-1: "they are quite coarse-grained and thus they simplify the analysis a lot"**
> >
> > **R3-1:** The reviewer raises a concern about estimation accuracy. We agree that current public disclosures contain substantial noise and incompleteness. In practice, it is extremely difficult, and often impossible, to know for any given model its exact training duration, hardware composition, model variants, or other configuration details that determine emissions. Under such constraints, producing perfectly granular reconstructions of model footprints is not feasible for any study.
> >
> > It is important to emphasize that this paper does not aim to exactly reconstruct the carbon footprint of each model. While we strive for the highest accuracy achievable, our goal is to provide conceptually correct and quantitatively reliable estimates whose uncertainty is well within one order of magnitude. This is the level of precision needed to resolve the community’s core uncertainty about scale. At present, it is unclear whether models associated with the Hugging Face ecosystem emit thousands, tens of thousands, or hundreds of thousands of tons of CO₂ during training. Our results deliver, for the first time, visibility at the correct scale and a clear, data-driven answer to this question. Absolute precision is unattainable with existing disclosures. But we emphasize again that **we have taken all feasible steps to ensure that our estimates are accurate, reasonable, and grounded in available information.**
> >
> > Additionally, our estimation framework is intentionally designed to be simple and executable at large scale. Any highly complicated framework would be hard to apply consistently across thousands of heterogeneous models. As our goal is to provide the community and industry with a higher-level, scalable perspective on carbon emissions. Some degree of methodological simplification is therefore not only unavoidable but necessary to make such large-scale estimation tractable and actionable.
> >
> > To provide an empirical assessment of estimation accuracy, we additionally evaluate the framework on models that *publicly disclose* their total training emissions. After filtering low-quality disclosures and applying a 95% symmetric trimming procedure to mitigate heavy-tailed outliers, the robust evaluation yields:
> > - **MAPE = 0.42**
> > - **Median RE = 0.32**
> > - **74% / 82%** of models lie within **×2 / ×3** of their disclosed values.
> >
> > These results provide a data-driven uncertainty envelope for the estimator. For HF-wide aggregates, this corresponds to an uncertainty of approximately **30–40%** for typical models. We have incorporated these empirical bounds and the full evaluation into Section 4.3 of the revised manuscript.
> >
> >
> > > **W3-2: "it is assumed that the GPU operates at maximum TDP"**
> >
> > **R3-2:** We argue that our framework is not based on a fixed power assumption in a way that would materially affect the results. The core idea is that for a given GPU architecture, the energy required to complete a fixed amount of computation can be reasonably estimated. Whether the GPU operates near its peak TDP or at a reduced power level, throughput scales in a similar way. If a GPU runs at roughly half of its maximum TDP, its effective throughput also decreases, which lengthens the training time. The total energy needed to finish the same number of floating point operations therefore stays relatively stable. In other words, while instantaneous power may fluctuate, the energy per unit of computation for a specific architecture remains within a predictable range.
> >
> > We also recognize that differences in system efficiency, such as pipeline optimization, dataloader performance, and communication overheads, can introduce additional variation. These factors do cause some deviation, but they remain far below one order of magnitude and do not affect our main goal, which is to understand the correct scale of emissions across the ecosystem. Our conclusions about the overall picture remain robust under realistic variation in GPU power behavior.

---

> > > ### Author Response · Authors · 2025-11-21
> > > **Response to Reviewer yezs (3/4)**
> > >
> > > > **W3-3: "do not see this as a sufficient contribution over existing guidelines and public data for ICLR"**
> > >
> > > **R3-3:** One of our central contribution is to show that the Hugging Face ecosystem constitutes a large scale, publicly accessible, and audit compatible empirical corpus for carbon accounting. This perspective has not appeared in prior work and enables, for the first time, systematic estimation across thousands of openly released models. Building on this insight, we develop a unified estimation pipeline that handles heterogeneous metadata, incomplete disclosures, and inconsistent logging practices. Existing tools cannot operate at this scale or under such variability, so establishing a robust process for recovering and standardizing model-level information is an essential methodological step.
> > >
> > > Crucially, answering the scale question itself is a meaningful contribution. Whether the emissions associated with these models are on the order of thousands, tens of thousands, or hundreds of thousands of tons fundamentally changes how the community interprets environmental impacts and frames future standards and policy discussions. We believe this quantitative visibility is valuable for both the ICLR community and the broader AI and sustainability fields.
> > >
> > > Looking ahead, we emphasize that this paper is one component of a broader effort that the entire community needs to pay attention to. A full understanding of AI’s lifecycle emissions will require many complementary studies. No single work can cover all dimensions.
> > >
> > > What matters is building a cumulative and coherent evidence base. Our work provides one such foundational piece by establishing a large-scale empirical dataset and a practical estimation procedure that future studies can extend or integrate with other perspectives. As reporting and transparency improve, the field will be able to combine these contributions into a more complete understanding of AI’s environmental footprint.
> > >
> > > > **Q1: "How does the scaling law approximation (line 259) capture training time?.... how is it captured in your analysis when GPU-hours are not reported?"**
> > >
> > > **R1:** In our framework, the scaling-law approximation refers to a widely accepted empirical relationship in large-scale language model training: as model size increases, the amount of training data (tokens) used in practice also increases, and as a result total training FLOPs increase accordingly. Our estimation on LLMs uses this accepted relationship. When a model's parameters and approximate training tokens are known or recoverable, total FLOPs follow directly from standard transformer training formulas. Thus the scaling-law approximation captures the empirical fact that bigger models imply larger datasets and proportionally larger compute, consistent with current practice across the ecosystem.
> > >
> > > In our experiments, training duration is handled as follows:
> > > - **When GPU-hours or electricity are disclosed**, emissions are computed directly via *power × time × PUE × EF*, so a longer run naturally yields proportionally higher emissions.
> > > - **When GPU-hours are not disclosed but FLOPs are known**, emissions scale with total FLOPs; runtime is reconstructed only as an intermediate quantity using hardware throughput.
> > > - **When neither GPU-hours nor FLOPs are disclosed**, FLOPs are estimated from *parameters × tokens* using the scaling-law approximation. More tokens/epochs/stages imply larger FLOPs and proportionally higher emissions.
> > > - In Eqs. (3)–(6), **A_time** is also introduced as a runtime-amplification factor capturing overhead  (communication, data loading, restarts). It is folded into emissions estimation and ATCI, not treated as a separate metric.
> > >
> > > This framework leads to an important property: “training time” is not treated as an independent variable. If the architecture, batch size, and hardware remain fixed, running “longer” necessarily means performing more steps or processing more tokens, which is operationally equivalent to more FLOPs. A "2× longer" run is effectively "2× the compute," not the same compute stretched over more hours.
> > >
> > > This is also why, once emissions are expressed as ATCI, wall-clock time cancels out. ATCI depends only on total compute performed, hardware energy efficiency, and regional carbon intensity. It measures the carbon cost per unit of training work, independent of calendar time.
> > >
> > >
> > > >**Q2: "Your estimate states 240 tCO2/e for Llama 3 / 3.1, but the Llama 3 repo on Hugging Face [3] states 2,290 tCO2/e. This discrepancy could be because the paper is only considering a single model out of the Llama family, but this should be clarified."**
> > >
> > > **R2:** Thanks for pointing this out. The discrepancy arises because our table reports the emissions for a single model (8B) in the family, whereas Meta’s public number aggregates **the entire family's pretraining**. We will update Table 2 to avoid confusion.

---

> > > > ### Author Response · Authors · 2025-11-21
> > > > **Response to Reviewer yezs (4/4)**
> > > >
> > > > > **Q3: "Table 2 lists the disclosed emissions for some model series, but doesn’t include the estimations?"**
> > > >
> > > > **R3:** Table 2 is intended to present officially disclosed emissions for several well-known model series, to illustrate the scale of emissions across different model families. Some of these models are closed-source commercial systems and do not have Hugging Face repositories, so they do not enter our estimation pipeline and therefore have no corresponding estimated values. We will clarify this in the table caption to avoid ambiguity. We also compare the disclosed and estimated emissions for several HF model series in Table 2 of the revised manuscript.
> > > >
> > > > > **Q4: "The types of accelerator families considered are not explicitly defined anywhere that I can find in the paper — the caption of Figure 2 suggests that NVIDIA A100/A800 and NVIDIA H100/H800 GPUs are considered. How exactly are jobs assigned to one of these families? For the Tier 1 data it sounds like this data is provided, but for Tier 2 it must be imputed, and even for Tier 1 data, a choice must be made if the model was trained using a different family of accelerator (e.g., AMD, Google TPU). How is this handled?"**
> > > >
> > > > **R4:** Thank you for pointing this out. In the implementation, accelerators are mapped to a small set of *canonical families* with associated peak TFLOPS, average power, and efficiency: NVIDIA A100 / A100-80GB / A100-64GB / A800, H100 / H200 / H800, V100, A40, A30, T4, L4,RTX 6000 ADA, AMD MI250X / MI300X, and GoogleTPU V2 / V3 / V4 / V5E / V5P. Assignment proceeds in our experiments as follows.
> > > >
> > > > **Tier-1 (disclosed hardware).** When model cards disclose training hardware type, we use these strings directly:
> > > >   - If training hardware is TPU, we parse the pod name (e.g., “v4-128”, “v3-8”) and map to a canonical TPU family (TPU V2/V3/V4/V5E/V5P). If the generation cannot be resolved, we default to **TPU V3** as a mid-range training accelerator.
> > > >  - Otherwise we treat the device as GPU and normalize it using regex rules, matching patterns such as `A100`, `A100 80GB`, `A800`, `H100`, `H800`, `V100`, `T4`, `L4`, `A40`, `A30`, `MI250X`, `MI300X`, or `RTX 6000 ADA`. The matched family is then used with specific peak TFLOPS, average power, and efficiency.
> > > >
> > > > **Tier-2 / Tier-3 (imputed hardware).**  When hardware metadata is incomplete:
> > > >   - If training hardware type indicates **TPU** but the pod string is ambiguous, we still assign **TPU V3** as a conservative default.
> > > >   - If the model is known to use **GPUs** (as illustraed in the readmes) but training hardware type cannot be resolved to a specific family, we fall back to **A100-class** assumptions (A100 peak TFLOPS and ~0.30–0.35 efficiency, 400 W power) as a representative datacenter GPU.
> > > >
> > > > AMD and TPU jobs are not collapsed into NVIDIA families: **MI250X** and **MI300X** have their own TFLOPS/power entries, and TPUs are handled via dedicated TPU families. Only when no reliable family can be inferred do we use an A100-class default for GPUs, or TPU V3 for TPUs, to keep assumptions conservative and consistent. We will add a brief description of this mapping (and the list of canonical families) to the Appendix A 8.1 in the revised manuscript.
> > > >
> > > > > **Q5: Minor comment — it may make sense to report CO2e/FLOP in a differently-scaled metric such as grams of CO2e/FLOP, since reporting in tons gives small values that are difficult to conceptualize (e.g., 10^{-22} on line 309).**
> > > >
> > > > **R5:** Thank you. This is a very helpful suggestion. We have switched to a more interpretable unit as ton CO₂e per EFLOP (ton/EFLOP)  in the revised manuscript.
> > > >
> > > > > **Q6: How is the grid region for carbon intensity chosen for each model? Is it purely based on the country of the model’s origin? Even within a single country, there can be a massive difference in the carbon intensity of the grid(s) — the US is a good example of this (e.g., compare California with Missouri).**
> > > >
> > > > **R6:** To assign regional carbon intensity, we use a sequential procedure designed to reflect the best available information.
> > > > - If a model card specifies the training region or reports a region-specific emission factor, we take that value directly.
> > > > - If this information is missing, we infer the region from the training organization’s compute infrastructure or institutional affiliations. This step is supported by an agent-based web search process that inspects each model’s public documentation and related organizational profiles.
> > > > - The Hugging Face author location is used only when neither of these sources is available.
> > > > We have clarified the complete procedure for assigning regional carbon intensity in the revised manuscript (Section 3.2).

---

> > > > > ### Comment · Reviewer_yezs · 2025-11-25
> > > > > **Reviewer Response**
> > > > >
> > > > > Dear authors,
> > > > >
> > > > > Thanks for your response.  My questions have largely been addressed.  I understand the challenges with existing disclosures and how that forces you to make some sensible assumptions to impute estimates.
> > > > >
> > > > > In your final version, please include these clarifications from the discussion phase about your methodology (in Appendix if necessary).  As reviewer bz5z mentioned, I think it will be important that the full workflow and implementation of the proposed estimation is made publicly available -- both for broader impact (enabling quick order-of-magnitude emissions estimates for any ML model on Hugging Face), and for sensitivity analysis (e.g., to see how including more information and/or changing assumptions would change estimates).
> > > > >
> > > > > Contingent on the above points, I will raise my score.

---

### Official Review · Reviewer_bz5z · 2025-10-31

**Soundness:** 2
**Presentation:** 2
**Contribution:** 3
**Rating:** 4
**Confidence:** 4

**Summary:**

The paper proposes a first step to quantify carbon emissions of open-weight models available on the Huggingface platform, based on their FLOPS consumed to train a released model checkpoint.  The calculation per model includes a conventional and fairly standard set of factors:  hardware (GPU family used to train) × efficiency × computation × system amplification × infrastructure × environment (regional energy efficiency information).  What’s novel in this paper is (1) aiming to account for a wide collection of models (moving beyond case studies on a single model or family, as in previous related work), starting with those that disclose the most information and with the most downloads, and expanding out; separating out the FLOPS (computation) used to train a model from all other factors, which are rolled into “ATCI” and empirically validated.  The paper culminates in numerical estimates for three tiers of Huggingface-available models, based on the amount of information for CO2 estimation that was disclosed, with Tier 1 models being used to calibrate and validate their estimates, while Tier 3 has most variables inferred (with downloads above 5K, 1K, and 100).   They also find how different model training routines (base or instruct models) and modalities (CV, NLP, and multimodal) impact the general CO2 training emission.

**Strengths:**

The paper is clearly written and motivated. As the authors say, “AI lacks a dedicated set of measurement practices, disclosure standards, and systematic methodologies for embodied carbon emissions.”  The paper is a first step toward a broadly (though not universally) applicable tool for estimating the footprint of AI.  It aims to help the field move from speculations and empty statements about environmental impact and actually measure it at scale.

Authors provide an empirical reference of the ATCI (AI training carbon intensity) metric, which allows for estimation of carbon emissions in past (and future?) training runs. The metric takes into account both hardware and regional energy characteristics and is the first to give us a general estimate of the impacts of model training on carbon emissions and contextualize it within other CO2-emitting human activities.

The authors seem to provide most of the necessary information/simplifications used to build the metric. Authors provide  explanations on how the data was collected, validated, and processed, as well as are committed to releasing it and their codebase so the community can build upon it.

**Weaknesses:**

The use of LMs with agents to extract information from repositories to fill in the details for all the models is interesting but hardly discussed.  Will these tools also be made publicly available so the process can be improved?

Although the comparison of CO2 emissions provided around line 87 helps, it would also be useful to have a ceiling for some of those comparisons, i.e., 2k model trainings were equivalent to around 7k cars annual emissions, but how many cars do we currently have in a country? And how long does it take to have 2097 new models on HuggingFace? Some of this information is in Figure 3 later in the text, but I feel I still don’t have a clear picture of how bad for the environment training is compared to other everyday activities and industries.

The estimation approach does not account at all for uncertainty propagation.  Each factor is simply estimated and they are multiplied together.  There’s not even an attempt to follow significant digit rules – e.g., Table 1a has 7 significant digits in the rightmost column.

The methodology for confirming that the estimates are valid is not detailed in the main paper.  I was unable to confirm that the approach was sound.  I’m actually unclear of whether I should view the differences within rows of Table 2 (estimation vs. disclosed) as error in the method or error in what’s disclosed or perhaps both.  The paper should be more clear about this.  On line 426, the authors mention that “estimation errors remain within an acceptable range” without actually determining what an acceptable range of error is and how different their estimates were from disclosed emissions in the text. The latter information can be inferred from Table 2, but it would be helpful to report aggregate values so we are more easily aware of how far off from the empirical values the estimate is. In addition, the authors make use of a lot of unified/aggregate values for the parameters in the metric, and it is not clear to me how much the errors would compound.

The stated assumption that the compute used to train a model comes from the same region as the author who uploaded the model seems very hard to support when estimating a regional electricity carbon intensity for each model.  Organizations that train models often use compute clusters in distant locales, driven by a concerns about price and environmental impact.

The models studied were trained at different times across five years.  Apart from specific hardware, there is no discussion of whether the factors needed to estimate ATCI are time-sensitive.  Perhaps cooling technology has become more efficient, for example.  It would be good to discuss this.

An undiscussed caveat with these estimates is that they consider (as far as I understand it) only the final model that is released.  But organizations that train models – especially those that do pretraining – carry out extensive experiments to choose hyperparameters, architecture details, data mixes, etc.  The final model is likely only a small fraction of the total compute used in the process.  I suspect that taking this into account would change the conclusion that “instruction models accumulate higher per-checkpoint emissions.”

Another concern I have that was addressed by the authors in the Limitations section is that models might emit much more carbon during their inference lifetime. Since the paper is concerned mostly with models with more than 5000 downloads, it is reasonable to expect that such models will have a high inference carbon emission. It would be interesting to have a rough estimate of how inference CO2 emissions would scale with usage, although I understand that measuring this is a challenging problem on its own.

It is acknowledged as a limitation that proprietary models are not included here.  From Table 2, where four non-HF models with disclosed emissions (not included in this paper’s estimates), we see that the footprint of training those models is likely orders of magnitude greater than what can be estimated through this paper’s methodology.  The models that can be accounted for, it seems, are contributing a tiny fraction to the real problem.


Typos:
Some citations are not in the correct format (lines 51, 106)
Extra “.” on line 106

**Questions:**

On line 38, the authors say that concerns about the environmental impact of AI remain conceptual. Later, on line 53, the authors place their contributions as a “conceptual estimate of its overall impact”. The term “conceptual” seems to mean loaded as it means different things in these two sentences. Is the proposed methodology also conceptual? A question that came up is: how often would the empirical parameters used in the ATCI have to be revisited?

---

> ### Author Response · Authors · 2025-11-21
> **Response to Reviewer bz5z (1/4)**
>
> Dear Reviewer bz5z,
>
> Thanks for your thoughtful comments and valuable suggestions. Our detailed responses are provided below.
>
> > **W1: "The use of LMs with agents to extract information is interesting ... Will these tools also be made publicly available so the process can be improved?"**
>
> **A1**: We appreciate the reviewer's recognition of our agent-based extraction pipeline. Both the tools and the data will be publicly released. We will make available (i) the full agent-assisted metadata extraction and emissions estimation pipeline, (ii) the Tier-2/Tier-3 regression models used for imputation, and (iii) the consolidated Hugging Face–wide metadata and emissions tables. This will allow others to audit, extend, or improve the workflow.
>
> > **W2: "it would also be useful to have a ceiling for some of those comparisons, i.e., 2k model trainings were equivalent to around 7k cars annual emissions, but how many cars do we currently have in a country? And how long does it take to have new models on HuggingFace? "**
>
> **A2:**  We thank the reviewer for this helpful suggestion. We have added ceiling comparisons in the revision to contextualize the environmental scale better. The estimated emissions are comparable to approximately 3 to 4 percent of the entire passenger car fleet of a small country such as Iceland. Hugging Face receives close to two thousand new high-download models within a typical period of ten to twelve months, and recent trends indicate that this number is increasing toward three to four thousand per year, so the ecosystem accumulates a similar amount of emissions roughly once per year.
>
> > **W3: The estimation approach does not account at all for uncertainty propagation. Each factor is simply estimated, and they are multiplied together. There’s not even an attempt to follow significant digit rules – e.g., Table 1a has 7 significant digits in the rightmost column.**
>
> **A3**: Thank you for raising this important point. In the revised manuscript, we have added a dedicated Section 3.5 on **"Uncertainty Propagation"** in the main paper and Appendix A.2 of **"Sources of Estimation Error Across Models"**, where we analyze how uncertainty enters through the multiplicative factors in Equation.3 - 6, apply first-order relative-error propagation in Equation.7, and report tier-specific theoretical uncertainty ranges.
>
> Additionally, we revised all numerical reporting to strictly follow the significant-digit rules. For both total emissions and ATCI, the number of significant digits now matches the precision justified by the least accurate input factor, and uncertainty bounds are rounded to one significant digit. ATCI values are therefore reported with only one or two significant digits, and aggregate emissions are presented at an "order-of-10^4" precision aligned with their propagated uncertainty.
>
> > **W4: "The methodology for confirming that the estimates are valid is not detailed in the main paper. The paper should be more clear about this. ... it would be helpful to report aggregate values so we are more easily aware of how far off from the empirical values the estimate is."**
>
> **A4**:We thank the reviewer for raising this point. To provide an empirical uncertainty estimate, we evaluate our framework on models that *publicly disclose* their total training emissions. After removing low-quality disclosures and applying a 95% symmetric trimming procedure to mitigate heavy-tailed outliers, the robust evaluation yields:
>
> - **MAPE = 0.42**
> - **Median Relative Error = 0.32**
> - **74% / 82%** of models fall within **×2 / ×3** of their disclosed values
>
> These results provide a data-driven error envelope for our estimator. For the HF-wide totals, this implies an uncertainty of roughly **30–40%** for typical models, with only a small number of extreme outliers beyond this range. We have included these empirical bounds and the full evaluation table in the Section 4.3 of revised manuscript to make the uncertainty range explicit.

---

> ### Author Response · Authors · 2025-11-21
> **Response to Reviewer bz5z (2/4)**
>
> > **W5: "In addition, the authors make use of a lot of unified/aggregate values for the parameters in the metric, and it is not clear to me how much the errors would compound."**
>
> **R5:** The parameters entering ATCI indeed come with uncertainty, and using aggregate values raises the question of error compounding. To address this, we conducted several quantitative analyses in the revision.
>
> First, we use **realistic perturbations** to the three key components (FLOPs ±30%, hardware/runtime/PUE ±20%, and grid emission factor ±10%) and 1,000 Monte Carlo samples per model, we estimate each factor’s contribution to total variance. The averaged variance shares are:
> - **FLOPs estimation:** ~66%
> - **Hardware/runtime/PUE:** ~27%
> - **Grid emission factor:** ~7%
> This breakdown shows that FLOPs estimation is the dominant source of uncertainty, hardware assumptions contribute moderately, and regional carbon intensity accounts for only a small share. This analysis is now included in Appendix A.3 of the revised manuscript.
>
> Furthermore, our **pseudo-missingness experiment** (see Appendix A.4) is conducted to explicitly quantify the uncertainty introduced by Tier 2 and Tier 3 estimation, which reflects the missing metadata disclosure patterns observed on Hugging Face.
> **1. Ground-truth selection**
> We use **all Tier 1 models** as high-confidence ground truth, including both those with direct energy disclosure or those with complete training metadata.
> **2. Constructing pseudo Tier 2 / Tier 3 samples**
> We randomly sample 70% of Tier 1 models and **deliberately mask metadata** to simulate realistic missingness:
> - **Pseudo Tier 2:** retain FLOPs, emission factor, and GPU family; mask hardware type, runtime, and direct/disclosed energy.
> - **Pseudo Tier 3:** further remove FLOPs, leaving only parameter count, emission factor, and GPU family.
>
> These masked models are re-evaluated using the **exact Tier 2 and Tier 3 regression pipelines** in the paper. Predicted emissions are compared with ground truth using relative error (RE). Results are summarized in this table:
>
> **Table: Pseudo-missingness experiment (Tier 2 and Tier 3 uncertainty)**
> | Pseudo Tier | n   | Median RE | P90 RE |
> |------------|-----|-------------|-------|
> | Tier 2 (FLOPs-based)   | 312 |  **0.57** | **1.20** |
> | Tier 3 (Params-based)  | 123 |  **0.99** | **1.92** |
>
> Tier-2 estimates remain highly stable, with a median RE of 0.57 and about 90% of predictions exhibiting ~1.2× relative error; Tier-3 estimates remain informative despite minimal metadata, with a median RE of 0.99 and about 90% of predictions exhibiting ~2× relative error. These results show that Tier 2/Tier 3 estimates are not exact but remain **predictive at the order-of-magnitude level under realistic missingness patterns**.
>
> > **W6: "The stated assumption that the compute used to train a model comes from the same region as the author who uploaded the model seems very hard to support when estimating a regional electricity carbon intensity for each model. Organizations that train models often use compute clusters in distant locales, driven by a concerns about price and environmental impact."**
>
> **R6:** To assign regional carbon intensity, we use a sequential procedure designed to reflect the best available information.
> - If a model card specifies the training region or reports a region-specific emission factor, we take that value directly.
> - If this information is missing, we infer the region from the training organization’s compute infrastructure or institutional affiliations. This step is supported by an agent-based web search process that inspects each model’s public documentation and related organizational profiles.
> - The Hugging Face author location is used only when neither of these sources is available.
> We have clarified the complete procedure for assigning regional carbon intensity in the revised manuscript (Section 3.2).
>
> We fully acknowledge that the ideal solution would be to use actual datacenter locations for each training run, but such information is not publicly available for the vast majority of models studied. Our framework is designed to incorporate more precise region metadata whenever it becomes available, and we will update the dataset accordingly in future releases.
>
> In the revised manuscript, we add a discussion noting that (i) PUE values for major cloud providers have improved gradually but remain within a narrow range (≈1.1–1.3), (ii) our sensitivity analysis shows that these variations translate to a modest impact on total emissions (~7%).

---

> > ### Author Response · Authors · 2025-11-21
> > **Response to Reviewer bz5z (3/4)**
> >
> > > **W7: "An undiscussed caveat with these estimates is that they consider (as far as I understand it) only the final model that is released. But organizations that train models – especially those that do pretraining – carry out extensive experiments to choose hyperparameters, architecture details, data mixes, etc. The final model is likely only a small fraction of the total compute used in the process. I suspect that taking this into account would change the conclusion that “instruction models accumulate higher per-checkpoint emissions."**
> >
> > **R7:** We thank the reviewer for raising this important point. Our estimates reflect only the compute associated with released training checkpoints. Intermediate experiments such as ablations, hyperparameter sweeps, and discarded runs are almost never documented at scale and therefore cannot be reconstructed from publicly available artifacts. As such, our estimates should be interpreted as lower bounds on project-level compute. We will clarify this in the revised manuscript.
> >
> > Crucially, this limitation does not reduce the value of quantifying the emissions of all observable training events, which constitute the only verifiable components of the open ecosystem and the only scope that can be assessed systematically today. We agree that estimating the “below-the-surface” R&D emissions is an important direction for future work, and may be approached through indirect signals such as data-center utilization statistics or compute-market capacity indicators. We view the present work as a necessary empirical foundation for these future extensions.
> >
> > > **W8: "It would be interesting to have a rough estimate of how inference CO2 emissions would scale with usage, although I understand that measuring this is a challenging problem on its own."**
> >
> > **R8**: We agree with the reviewer’s understanding of the broader scope of AI-related carbon emissions. We share the same view: a comprehensive industry-scale assessment must ultimately integrate many components beyond open-source model training, including inference workloads, industrial deployments, and the full lifecycle impacts of GPU manufacturing.
> >
> > In principle, the same FLOPs-based framework can be extended to inference: inference emissions can also be modeled from per-request FLOPs, hardware efficiency, and the corresponding carbon intensity. However, unlike training—where FLOPs and hardware configurations are usually fixed—accurate inference estimation requires detailed usage data such as request volume, model deployment settings, batching behavior, and accelerator utilization. These data are rarely disclosed or consistently available across models or platforms.
> >
> > At the same time, incorporating these dimensions requires fundamentally different data sources and methodologies. As reflected in our title, this work focuses specifically on training-related emissions, which is a foundational and tractable component of the larger picture. Our long-term goal is to extend the analysis to the additional areas mentioned above and combine heterogeneous datasets to build a truly holistic industry-level view. We will add a brief discussion of this extension in the revised manuscript.

---

> > > ### Author Response · Authors · 2025-11-21
> > > **Response to Reviewer bz5z (4/4)**
> > >
> > > > **W9: "proprietary models are not included here. From Table 2, where four non-HF models with disclosed emissions (not included in this paper’s estimates), we see that the footprint of training those models is likely orders of magnitude greater than what can be estimated through this paper’s methodology. The models that can be accounted for, it seems, are contributing a tiny fraction to the real problem."**
> > >
> > > **R9:** We fully agree that some proprietary frontier models (e.g., GPT-4, Gemini) have very large training footprints, and our study does not attempt to estimate emissions for these closed-source systems. Only a very small number of proprietary models have ever disclosed any emissions, which limits systematic analysis in that domain.
> > >
> > > However, we respectfully clarify that the open-source ecosystem we study is far from a “tiny fraction.” Hugging Face hosts the **majority of openly released foundation, instruction, and multimodal models worldwide**, representing thousands of checkpoints and tens of thousands of downstream fine-tuning runs.
> > > - While individual HF models are smaller than frontier proprietary models, the **cumulative footprint is substantial**: HF-wide training emissions reach **tens of thousands of tons of CO₂e** for downloads > 5k, and **hundreds of thousands of tons** for downloads > 100.
> > > - The open-source ecosystem is growing **exponentially**. Even if proprietary frontier models dominate the very top end of compute, the rapidly expanding volume of open models forms a **large-base, long-tail distribution** whose aggregate environmental impact cannot be ignored.
> > >
> > > Our contribution is therefore complementary to analyses of proprietary frontier models. By quantifying emissions across the broad open-source landscape rather than focusing only on a few extreme training runs, we provide a more complete picture of ecosystem-wide AI training emissions. We will revise the manuscript to articulate this complementarity more clearly.
> > >
> > > > **W10: Some citations are not in the correct format (lines 51, 106) Extra “.” on line 106**
> > >
> > > **A1:** Thank you for pointing this out. We appreciate the careful reading. We have corrected the citation formatting issues in the revised manuscript.
> > >
> > >
> > > > **Q1: On line 38, the authors say that concerns about the environmental impact of AI remain conceptual. Later, on line 53, the authors place their contributions as a “conceptual estimate of its overall impact”. The term “conceptual” seems to mean loaded as it means different things in these two sentences. Is the proposed methodology also conceptual? A question that came up is: how often would the empirical parameters used in the ATCI have to be revisited?**
> > >
> > > **R1:** Thank you for the helpful observation. In the introduction, “conceptual concern” refers to the *public discourse* around AI sustainability, which often lacks quantitative grounding. In contrast, our “conceptual estimate” refers to providing the *first system-level,order-of-magnitude quantification* of HF-wide emissions. The methodology itself is not conceptual or hypothetical: it is fully empirical, grounded in publicly available metadata, FLOPs-based estimation, and regression models calibrated on Tier-1 disclosures.
> > >
> > > Regarding parameter updates, we note that ATCI combines quantities that evolve slowly—GPU efficiency and power change only with new hardware generations; PUE for major providers has remained in the ~1.1–1.3 range for years; and regional emission factors are updated annually. Importantly, in our study we compute the **training emissions for each model using the parameters of its training year**.
> > >
> > > The estimates already reflect the approximate temporal context of the model’s training period, and there is **no need to retroactively update ATCI for past models**. Only if one aims to construct a *sector-level* or *future-facing* “industry-wide ATCI” for the following years would periodic updates be appropriate.
> > >
> > > Thus, ATCI does not require continuous revision. Updating it *annually* or *per major hardware generation* is sufficient. We will clarify this distinction and the update frequency for ATCI in the revision. Overall, our framework is empirical rather than conceptual, while some elements of the broader sustainability discussion remain conceptual due to the lack of public data.

---

> > > > ### Author Response · Authors · 2025-11-27
> > > > **Follow-up Before the Discussion Period Ends**
> > > >
> > > > Dear Reviewer bz5z,
> > > >
> > > > With the discussion phase coming to a close, we just wanted to gently follow up. All responses have been posted, and the revised manuscript has been uploaded. We are glad to clarify any remaining questions or comments you might have.
> > > >
> > > > Thank you very much for your time and constructive feedback.
> > > >
> > > > Best regards,
> > > > Authors

---

### Official Review · Reviewer_VGTc · 2025-10-31

**Soundness:** 3
**Presentation:** 3
**Contribution:** 3
**Rating:** 6
**Confidence:** 4

**Summary:**

This is a FLOPs-based carbon emission estimation framework for LLM training by approximating the emissions of open source models on Huggingface. The key idea is to first approximate the total computational cost in FLOPs required to train a given model.
This quantity is then converted into energy consumption based on the efficiency characteristics of the hardware likely used for training and finally into carbon emissions by applying the carbon intensity of electricity in the relevant region. This conversion can be interpreted as assigning a training carbon intensity (ATCI), which means training emissions per FLOP, which reflects both hardware energy efficiency and regional energy mix.

Training emissions can be calculated by hardware × efficiency × computation × system amplification × infrastructure × environment. The authors adopt a three-tier framework to include models having less/more training information in documentation. the models having rich disclosure (gpu hrs, hardware types, flops), partial (flops), minimal info (parameters).
Results show mainly that CV and multimodal exhibit higher training emission intensity than NLP.  Instruction-tuned models show higher median model-level emissions than pretrained models.

**Strengths:**

The paper addresses an important and timely problem: quantifying AI’s environmental cost at scale.

Introduces the ATCI (AI Training Carbon Intensity) metric, a practical abstraction for emissions per FLOP.

The three-tier framework is systematic and allows estimation even when disclosure quality varies.

Provides an industry-scale analysis of thousands of models, revealing useful cross-domain insights (e.g., CV vs NLP, base vs instruction-tuned).

**Weaknesses:**

1. Paper focuses only on the training related carbon emissions and omit the inference related emissions. But it is important to model the inference emissions. Can this framework be used for inference estimates? I acknowledge that the authors mentioned this in limitations and future work.
2. Also while basing off these experiments only on HF models is good, we don't have an estimate of emissions of proprietary models. They have a three-tier framework in which only parameter count is considered in third tier, given that can we estimate emissions for propretary LMs?
3. Mainly two results are derived. One of which is: Instruction-tuned models show higher median model-level emissions than base pretrained models. Can we draw comparison across different LLM families?
4. How their approach is better/compare to [1] This paper also does flops based end-to-end carbon emission modeling.
5. Can this framework be used to model fine-tuning carbon emissions? Because once the models are trained and released on HF, fine-tuning and inference is done most often, so these emissions should be accounted for as well.

[1] Faiz, A., Kaneda, S., Wang, R., Osi, R., Sharma, P., Chen, F. and Jiang, L., 2023. Llmcarbon: Modeling the end-to-end carbon footprint of large language models. arXiv preprint arXiv:2309.14393.

**Questions:**

1.	What is the uncertainty or error margin of the reported totals (e.g., 33,446 t CO₂e)? Can they provide confidence intervals or upper/lower bounds?
	2.	In Tier 2 and 3, what representative hardware assumptions (GPU type, utilization, PUE) were used? How sensitive are results to those?
	3.	How were training regions determined for each model? If based on author affiliation, this may misrepresent actual data-center locations.
	4.	For Tier 3 regression (parameter-only), how accurate is the fit when tested on known cases?
	5.	Since the work relies heavily on imputed data, can the authors release their dataset and code for transparency and reproducibility?
	6.	The related work section omits LLMCarbon (Faiz et al., 2023) and other recent carbon-accounting tools (CodeCarbon, CarbonTracker). How does this framework differ from or improve upon them?
	7.	Can the ATCI metric be extended to inference or fine-tuning phases, and what would change in the estimation equation?

---

> ### Author Response · Authors · 2025-11-21
> **Response to Reviewer VGTc (1/3)**
>
> Dear Reviewer VGTc,
>
> We sincerely appreciate your valuable comments and suggestions. Our point-by-point responses follow below.
>
> > **W1. "...it is important to model the inference emissions. Can this framework be used for inference estimates..."**
>
> **A1:**
> We appreciate the reviewer’s point. We share the same view: A comprehensive assessment of AI’s environmental footprint ultimately needs to incorporate many components beyond training, including inference workloads, industrial deployments, and the lifecycle impacts of GPU manufacturing.
>
> Our FLOPs-based methodology is conceptually extensible to inference, because inference emissions can also be modeled from per-request FLOPs, hardware efficiency, and regional carbon intensity. The main challenge is data availability: unlike training, where compute budgets and hardware configurations are usually fixed, inference estimation requires detailed usage statistics such as request volume, batching behavior, deployment settings, and accelerator utilization. These data are rarely disclosed or consistently reported.
>
> For this reason, the present work focuses on training emissions, where the underlying computational workload is well defined and can be estimated systematically. Incorporating real-world inference telemetry remains an important direction for our future work.
>
>
> > **W2: "while basing off these experiments only on HF models is good, we don't have an estimate of emissions of proprietary models. They have a three-tier framework in which only parameter count is considered in third tier, given that can we estimate emissions for propretary LMs?"**
>
> **A2:** Thank you for the question. Hugging Face is currently the only ecosystem with a large number of publicly released model cards, making it the most complete and internally consistent source for cross-model validation.
>
> Proprietary LMs do not provide a curated or machine-readable corpus of training metadata, which makes systematic cross-checking or large-scale validation infeasible. Only a handful of proprietary LMs provide any training-related disclosures. Major closed-source systems such as Claude do not self-report training emissions, FLOPs, or GPU-hours, which limits the feasibility of systematic comparison at this stage. In this sense, HF offers the best available testbed for open, reproducible carbon accounting.
>
> Regarding Tier 3, it is not "parameter-only estimation." Tier 3 is used when FLOPs are not disclosed, but the estimation still leverages some metadata signals, including **parameter count, GPU type, regional carbon intensity, model modality, and accelerator family**. Parameter count serves as a statistical predictor of computational cost, while the other factors account for hardware efficiency and regional emissions. This allows Tier 3 to provide coarse yet meaningful emission estimates even when FLOPs are unavailable.
>
> We will also consider extending our dataset by incorporating any available disclosures from proprietary LMs as they become public, enabling future cross-ecosystem comparisons.
>
> > **W3: "Can we draw comparison across different LLM families?"**
>
> **A3**: Thank you for the question. We can draw comparison across different LLM families. In the revised manuscript, we also report family-level statistics for several widely used LMs. For each family, we aggregate the total training emissions. We have added some corresponding results to Appendix A.8 (Figure 7, 8) in the revised manuscript.
>
> | LM Family | Number of Models | Total Emissions (1,000 ton) |
> |-----------|---------|--------------------------|
> | Llama     | 202     | 13                       |
> | Qwen      | 134     | 7                        |
> | DeepSeek  | 34      | 2                        |
> | GLM       | 14      | 1                        |
> | Gemma     | 22      | 0.8                      |
>
> > **W4: "How their approach is better/compare to [1] (LLMCarbon). This paper also does flops based end-to-end carbon emission modeling."**
>
> **A4:** The goals, assumptions, and methodological scope of LLMCarbon and ours are fundamentally different. LLMCarbon infers energy use from detailed hardware and parallelism configurations, and validates its model on a small set of fully-specified LLMs. Our framework also converts compute to energy using hardware characteristics, but it is explicitly designed for ecosystem-level estimation under severe metadata sparsity.
>
> Crucially, instead of assuming fixed hardware efficiency or parallelism schedules, we empirically validate the FLOPs–emissions relationship through empirical regressions across hundreds of disclosed Tier-1 models. This allows our method to scale across thousands of heterogeneous NLP, CV, and multimodal models, whereas LLMCarbon applies to LLMs. A detailed comparison has been added in the revised manuscript.

---

> > ### Author Response · Authors · 2025-11-21
> > **Response to Reviewer VGTc (2/3)**
> >
> > > **W5: "Can this framework be used to model fine-tuning carbon emissions? Because once the models are trained and released on HF, fine-tuning and inference is done most often, so these emissions should be accounted for as well."**
> >
> > **A5:** Yes. Our framework naturally supports fine-tuning emissions. Each fine-tuned model is treated as an independent training run, and its emissions are estimated using the same FLOPs-based method as pretraining. If fine-tuning tokens are disclosed, FLOPs are computed directly; otherwise, we apply standard heuristics for SFT/RLHF workloads.
> >
> > We also use Hugging Face’s model-tree metadata to identify downstream checkpoints, ensuring that every fine-tuning run is counted as a separate emission event.
> >
> > >  **Q1: "What is the uncertainty or error margin of the reported totals? Can they provide confidence intervals or upper/lower bounds?"**
> >
> > **R1**: We thank the reviewer for raising this point. To provide an empirical uncertainty estimate, we evaluate our framework on models that *publicly disclose* their total training emissions. After removing low-quality disclosures and applying a 95% symmetric trimming procedure to mitigate heavy-tailed outliers, the robust evaluation yields:
> > - **MAPE = 0.42**
> > - **Median Relative Error = 0.32**
> > - **74% / 82%** of models fall within **×2 / ×3 ** of their disclosed values
> >
> > These results provide an error envelope for our estimator. For the HF-wide totals, this implies an uncertainty of roughly **30–40%** for typical models, with only a small number of extreme outliers beyond this range. We have included these empirical bounds in Section 4.3 of the revised manuscript to make the uncertainty range explicit. Our reporting results follow standard significant‐digit rules and consider uncertainty propagation, consistent with the propagated error in Equation 7 (See details in Appendix A.2).
> >
> > >  **Q2: "In Tier 2 and 3, what representative hardware assumptions (GPU type, utilization, PUE) were used? How sensitive are results to those?"**
> >
> > **R2**: Below we clarify (i) what hardware information we actually observe for Tier-2 and Tier-3 models, (ii) the representative hardware assumptions used when information is missing, and (iii) how sensitive our results are to those assumptions.
> >
> > **1) In our HF-wide dataset, there are 2,623 Tier-2 and 790 Tier-3 models:**
> > - Tier 2: 21% (547/2,623) report the hardware type (e.g., "GPU/TPU"), and 8.7% (228/2,623) specify a concrete accelerator model.
> > - Tier 3: 29% (227/790) report the hardware type, and 9.6% (76/790) specify a concrete accelerator model.
> > Among the Tier-2/3 models that do specify a GPU, roughly ~190 models fall into A100/H100/H800-class accelerators, and ~40–50 models use RTX 30/40-series or A6000/L40S-class GPUs,
> > with the remainder in "unspecified" GPUs.
> >
> > **2) Representative utilization and PUE assumptions**
> > When training hardware is missing, our FLOPs–regression pipeline defaults to the A-family (A100/A800-class) GPU group. The estimator therefore applies their representative characteristics (peak ≈312 TFLOPs, ~400 W power, efficiency ≈0.30, PUE = 1.2). This avoids mapping models to low-power consumer GPUs and keeps imputed cases calibrated to the same distribution as models with full metadata. We will make this fallback rule explicit in the revised manuscript.
> >
> > **3) Sensitivity of results to hardware assumptions**
> > To quantify uncertainty, we use realistic perturbations to the three key components (hardware/runtime/PUE ±20%, and grid emission factor ±10%) and 1,000 Monte Carlo samples per model. We estimate each factor’s contribution to total variance. The averaged variance shares are:
> > - **Hardware/runtime/PUE:** ~27%
> >
> > This breakdown shows that hardware assumptions contribute moderately. The quantitative analysis is included in Appendix A.3 of the revised manuscript.
> >
> > >  **Q3: "How were training regions determined for each model? If based on author affiliation, this may misrepresent actual data-center locations."**
> >
> > **R3:** Thank you for the question. Our procedure is hierarchical: **(i)** when a model card reports the training region or actual data-center locations, we use that value directly; **(ii)** When the region is not disclosed, we infer it based on the corresponding institution. We also evaluate the effect of uncertainty in regional emission factors and find that substituting reasonable neighboring regions generally changes the estimated emissions by no more than 7 percent. We will clarify this in the revised manuscript.

---

> > > ### Author Response · Authors · 2025-11-21
> > > **Response to Reviewer VGTc (3/3)**
> > >
> > > > **Q4: For Tier 3 regression (parameter-only), how accurate is the fit when tested on known cases?**
> > >
> > > **R4**: Thank you for the question. Because Tier-3 models do not have ground-truth emissions or energy consumption information (otherwise they would be Tier-1), we evaluate Tier-3 accuracy through a *pseudo-missingness* experiment on Tier-1 models. We take high-confidence Tier-1 samples, mask all FLOPs and hardware information, and re-estimate emissions using the exact Tier-3 (parameter-only) regression pipeline. This provides a controlled test of how well Tier-3 performs when only parameter count, GPU family, and region are available.
> > >
> > > The results show that Tier-3 remains informative despite minimal metadata: **Median RE ≈ 0.99** and **P90 RE ≈ 1.92** on 123 masked models. In other words, half of the Tier-3 reconstructions deviate from ground truth by less than ~100%, and 90% fall within roughly **2–3x**, depending on model. This experiment demonstrates that the Tier-3 regression retains meaningful predictive power even under severe metadata sparsity. We have included the results of this pseudo-missingness experiment in Appendix A.4.
> > >
> > > > **Q5: Since the work relies heavily on imputed data, can the authors release their dataset and code for transparency and reproducibility?**
> > >
> > > **R5**: We fully agree with the reviewer on the importance of transparency. Code, data-processing scripts, and the final imputed dataset used in this study will be released publicly. This includes (i) the full emissions estimation pipeline, (ii) the regression models for Tier-2 and Tier-3 imputations, and (iii) the cleaned Hugging Face–wide metadata tables. This will allow complete reproducibility of all results in the paper.
> > >
> > > > **Q6: The related work section omits LLMCarbon (Faiz et al., 2023) and other recent carbon-accounting tools (CodeCarbon, CarbonTracker). How does this framework differ from or improve upon them?**
> > >
> > > **R6**: Whereas LLMCarbon offers detailed estimates for a single LLM with complete inputs, our framework works with partial data and automates estimates for hundreds of models. Unlike CodeCarbon or CarbonTracker (as discussed in the literature review), which monitor live training runs, our method can retrospectively estimate emissions for models by leveraging their known properties (parameters, architecture) and reasonable assumptions for unknowns. This enables large-scale comparisons and reporting that the existing tools cannot easily support.  A detailed comparison has been added in the revised manuscript.
> > >
> > > > **Q7: Can the ATCI metric be extended to inference or fine-tuning phases, and what would change in the estimation equation?**
> > >
> > > **R7**:Yes. ATCI can be extended to both fine-tuning and inference, because its meaning (carbon cost per compute) stays the same. Only the phase-specific inputs change.
> > > - **Fine-tuning**: use the tokens, hardware information, and region/PUE of the fine-tuning setup.
> > > - **Inference**: use the characteristics of the inference hardware and the per-request FLOPs (e.g., FLOPs per generated token).
> > >
> > > In a word, ATCI itself does not change. Only the parameters plugged into it differ across phases.

---

> ### Author Response · Authors · 2025-11-27
> **Follow-up Before the Discussion Period Ends**
>
> Dear Reviewer VGTc,
>
> As the discussion period is approaching its end, we would like to kindly follow up. We have provided all our responses and uploaded the revised manuscript. We are fully available to address any remaining questions or concerns you may have.
>
> We truly appreciate your time and thoughtful consideration.
>
> Best regards,
> Authors

---

### Official Review · Reviewer_cMNg · 2025-11-01

**Soundness:** 3
**Presentation:** 3
**Contribution:** 2
**Rating:** 4
**Confidence:** 3

**Summary:**

This paper develops a structured approach to estimate the training carbon emissions of open-source AI models on Hugging Face. To address the current lack of consistent accounting standards in AI, the authors design a FLOPs-based estimation framework that links model training compute to carbon output through hardware efficiency and regional electricity carbon intensity. They introduce the AI Training Carbon Intensity (ATCI) metric—measuring CO₂-equivalent emissions per FLOP—to provide a practical and comparable way to estimate emissions when complete training data are unavailable. To handle uneven metadata quality, the study applies a three-tier classification system (Tier 1: detailed; Tier 2: partial; Tier 3: minimal) and combines automated LLM-assisted metadata extraction with manual cross-checking for verification.

**Strengths:**

- The introduction of the FLOPs-based estimation framework combined with the AI Training Carbon Intensity (ATCI) metric offers a practical, scalable approach to estimate emissions—even when disclosure is incomplete or absent.
- In contrast to earlier research centered on a few individual models, this study examines open-source Hugging Face models at an industry-wide scale. The extensive coverage enables a comprehensive estimate of the cumulative training emissions within the open-source AI ecosystem, offering a broader and more representative view than case-specific analyses.
- This paper presents comprehensive experimental results and systematically compares NLP, CV, and multimodal models. The analysis shows that vision workloads exhibit higher ATCI values, largely due to data-processing and pipeline overhead. It also differentiates between base pretraining and instruction tuning, revealing the often-overlooked cumulative carbon cost associated with repeated fine-tuning and alignment cycles.
- The paper is well written, with clear organization and logical flow throughout. In particular, the results are effectively visualized—figures and tables are well designed, easy to interpret, and enhance the overall clarity and impact of the presentation.

**Weaknesses:**

- The novelty of this paper is not enough. Even this papers mentation this is the first systematic accounting of training related carbon emissions for mainstream models hosted on Hugging Face. But the core methodology (FLOPs × carbon intensity) is not new—it has been previously proposed in works such as [1]-[5]. The novelty is primarily scale and aggregation, not conceptual framework.
- As the paper notes, its analysis focuses solely on training-related emissions while excluding other important sources such as inference, research, and full lifecycle impacts. This limitation reduces the completeness of the overall environmental assessment. It would be valuable for future work to extend the framework to include inference and other downstream processes, providing a more holistic view of AI’s total carbon footprint.
- It is a thoughtful approach that the paper classifies models into three tiers based on disclosure completeness. However, for Tier 2 and Tier 3 models, where key information is missing, the heavy reliance on estimation and regression methods—though practical—introduces considerable uncertainty into the results and may affect the precision of the overall emission estimates.
- In the Training FLOPs Estimation section, the study relies on heuristic approximations for different model architectures. Since model structures and training configurations vary widely, this approach may lead to estimation biases. It would strengthen the work to validate these approximation formulas using available disclosed training logs or verified benchmarks where possible, thereby improving the accuracy and credibility of the computed FLOPs values.
- The study provides limited ground-truth validation, relying on comparisons with only a small number of disclosed models (e.g., BLOOM, LLaMA) as shown in Table 2. While these examples demonstrate reasonable consistency, the validation set is too narrow to thoroughly evaluate the accuracy and robustness of the proposed estimation framework. Future work could expand the validation dataset by incorporating more models with partial disclosures, such as reported FLOPs or runtime statistics. Additionally, collaborating with model developers to access authentic training logs would provide stronger empirical grounding and improve the reliability of the framework’s estimates.

## References
**[1]** Strubell, Emma, Ananya Ganesh, and Andrew McCallum. "Energy and policy considerations for modern deep learning research." Proceedings of the AAAI conference on artificial intelligence. Vol. 34. No. 09. 2020.

**[2]** Patterson, David, et al. "Carbon emissions and large neural network training." arXiv preprint arXiv:2104.10350 (2021).

**[3]** Anthony, Lasse F. Wolff, Benjamin Kanding, and Raghavendra Selvan. "Carbontracker: Tracking and predicting the carbon footprint of training deep learning models." arXiv preprint arXiv:2007.03051 (2020).

**[4]** Lacoste, Alexandre, et al. "Quantifying the carbon emissions of machine learning." arXiv preprint arXiv:1910.09700 (2019).

**[5]** Luccioni, Alexandra Sasha, Sylvain Viguier, and Anne-Laure Ligozat. "Estimating the carbon footprint of bloom, a 176b parameter language model." Journal of machine learning research 24.253 (2023): 1-15.

**Questions:**

- Table 2 lists a few models that disclose their training emissions. Could the authors provide more detailed comparison data—such as the disclosed FLOPs, GPU-hours, or runtime statistics—for these models? Additionally, would it be possible to calculate and report the relative errors between the disclosed emissions and the framework’s predicted values to better assess the accuracy of the estimation method?
- Could the authors provide a quantitative breakdown of the overall emissions uncertainty, distinguishing the respective contributions from FLOPs estimation, hardware efficiency assumptions, and regional grid carbon intensity?

---

> ### Author Response · Authors · 2025-11-21
> **Response to Reviewer cMNg (1/4)**
>
> Dear Reviewer cMNg,
>
> Thank you for your insightful suggestions and comments. Below are our detailed responses.
>
> **W1**: "The novelty is primarily scale and aggregation, not conceptual framework."
>
> **A1:**
> We would like to offer a different perspective regarding the question of novelty. Our contribution goes beyond scale or aggregation and introduces several substantive and original advances.
> - **First, our work identifies that the Hugging Face model ecosystem itself can serve as a meaningful and analyzable target for carbon accounting.** This observation has not appeared in prior literature, and recognizing Hugging Face as a tractable, large-scale, and publicly accessible corpus for carbon assessment is a novel insight in its own right. Building on this, we introduce the first auditable and systematic assessment of its training-related emissions, providing macro-level visibility that the community previously lacked.
>
> - **Second, conducting an accounting of the eligible Hugging Face models is technically non-trivial**. The models vary widely in metadata quality, disclosure completeness, training configurations, and logging formats. Our study introduces a unified estimation pipeline together with data-recovery and harmonization procedures that make large-scale, cross-model carbon estimation feasible. This unified methodological framework is itself a contribution because it enables analyses that would not be possible with existing tools.
>
> - **This work provides a methodology for estimating carbon emissions at this scale.** Our approach leverages training FLOPs as a unifying variable for estimating carbon emissions, enabling consistent comparisons across diverse architectures and training setups. Such FLOPs-centered methodology is not present in the references mentioned by the reviewer, and it plays a central role in making our large-scale quantification feasible.
>
> - **We also contribute the methodology and empirical evidence.** FLOPs × EF is only the final step; the real challenge is estimating FLOPs, hardware, region, PUE, and runtime at scale. We address this via architecture-based FLOPs derivation and runtime backsolving (Eq. 4–5), and validate the FLOPs–emission relationship through large-scale regressions (Figure 2, Table 10). We further propose AI Training Carbon Intensity (ATCI; Eq. 6) as a standardized, interpretable efficiency metric aligned with established environmental accounting frameworks.
>
> - **Finally, scale and aggregation are not mere engineering details**; they constitute a substantive contribution to our community. Without such work, the field would lack the foundational knowledge needed to understand AI’s environmental impacts at this scale. Large-scale empirical mapping efforts are a well-established form of novelty in many scientific domains.
>
> For these reasons, we believe the contribution of this paper is both novel and important.
>
>
> **W2**. “...the core methodology (FLOPs × carbon intensity) is not new...it has been previously proposed in works such as [1]-[5]”
> **A2:**
> The cited papers or the FLOPs × intensity idea **can not undermine the novelty of our work**. The cited works do not treat FLOPs as a principled foundation for carbon accounting, nor do they develop a FLOPs-based attribution framework in the way our work does.
> - **[1] Strubell et al.** do not involve FLOPs; emissions are computed solely as measured/reproduced electricity × regional EF for several NLP models (GPT-2, BERT,etc).
> - **[2] Patterson et al.** estimate FLOPs for Google models (T5, Meena,etc), but emissions are still derived only from measured electricity × regional EF, not FLOPs-based estimation.
> - **[3] Anthony et al.** and **[4] Lacoste et al.** use FLOPs as a proxy for electricity consumption, without analyzing emissions-per-FLOP or cross-model carbon intensity.
> - **[2], [3], [4]** consider hardware efficiency (FLOP/s), but none treat FLOPs as a standardized or comparable metric for carbon efficiency of AI models.
> - **[5] Luccioni et al.** compute BLOOM’s emissions from internal energy logs and regional EF, without FLOPs estimation or FLOPs-based intensity metrics.
> - In addition, prior work either focuses on single-model or single-architecture case studies ([1],[5]), depends on complete metadata or internal telemetry ([2], [5]), or provides experiment-level monitoring tools ([3], [4]). In contrast, our framework scales to thousands of models and enables reproducible, platform-wide carbon attribution.
> - Meanwhile, emissions per FLOP is only the final arithmetic step. The key bottleneck, overlooked in [1]–[5], lies in estimating FLOPs, hardware, region, PUE, and runtime for thousands of heterogeneous HF models with missing disclosures. Our emission equations (Eq. 4–5) formalize this relationship, and large-scale regressions (Appendix 8.2–8.3) empirically validate it. **We therefore provide both the theoretical and empirical foundations.**

---

> ### Author Response · Authors · 2025-11-21
> **Response to Reviewer cMNg (2/4)**
>
> **W3**:"....It would be valuable for future work to extend the framework to include inference and other downstream processes..."
>
> **A3**:
> We appreciate the reviewer's understanding of the broader scope of AI-related carbon emissions. We share the same view: a comprehensive industry-scale assessment must ultimately integrate many components beyond open-source model training, including inference workloads, industrial deployments, and the full lifecycle impacts of GPU manufacturing.
>
> At the same time, incorporating these dimensions requires fundamentally different data sources and methodologies. As reflected in our title, this work focuses specifically on training-related emissions, which is a foundational and tractable component of the larger picture. Our long-term goal is to extend the analysis to the additional areas mentioned above and combine heterogeneous datasets to build a truly holistic industry-level view.
>
> We therefore do not consider this focus a weakness, but rather evidence of the value of our contribution: it provides an essential and previously missing piece of a much larger landscape, without which the full picture cannot be constructed.
>
> **W4**: "For Tier 2 and Tier 3 models, where key information is missing, the heavy reliance on estimation and regression methods reliance on estimation and regression methods—though practical—introduces considerable uncertainty....affect the precision of emission estimates..."
>
> **A4:** We agree that Tier 2 and Tier 3 estimates inevitably involve uncertainty because these models lack key metadata. This uncertainty reflects an inherent characteristic of the open-source ecosystem rather than a limitation of our method. Our contribution is precisely to make these otherwise unanalyzable cases measurable rather than discarding them.
>
> To explicitly quantify the uncertainty introduced by Tier 2 and Tier 3 estimation, we conduct the following **pseudo-missingness experiment** in Appendix A.4, which reflects the missing metadata disclosure patterns observed on Hugging Face.
>
> **1. Ground-truth selection:** We use **all Tier 1 models** as high-confidence ground truth, including both those with direct energy disclosure or those with complete training metadata.
>
> **2. Constructing pseudo Tier 2 / Tier 3 samples:** We randomly sample 70% of Tier 1 models and **deliberately mask metadata** to simulate realistic missingness:
> - **Pseudo Tier 2:** retain FLOPs, emission factor, and GPU family; mask hardware type, runtime, and direct/disclosed energy.
> - **Pseudo Tier 3:** further remove FLOPs, leaving only parameter count, emission factor, and GPU family.
>
> These masked models are re-evaluated using the **exact Tier 2 and Tier 3 regression pipelines**. Predicted emissions are compared with ground truth using relative error (RE). Results are summarized in this table:
>
> **Table.1  Pseudo-missingness experiment (Tier 2 and Tier 3 uncertainty)**
> | Pseudo Tier | n   | Median RE | P90 RE |
> |------------|-----|-------------|-------|
> | Tier 2 (FLOPs-based)   | 312 | **0.57** | **1.20** |
> | Tier 3 (Params-based)  | 123 | **0.99** | **1.92** |
>
> - **Tier 2 estimates remain highly stable.** Median RE ≈ **0.57**; **90%** of predictions fall within **~1.2×** of ground truth.
> - **Tier 3 remains informative despite minimal metadata.** Median RE ≈ **0.99**; **90%** of predictions fall within **~2×**.
>
> These results show that Tier 2/Tier 3 estimates are not exact but remain **predictive at the order-of-magnitude level under realistic missingness patterns**.
>
> **W5**:"....provides limited ground-truth validation, relying on comparisons with only a small number of disclosed models as shown in Table 2...."
>
> **A5:**
> We appreciate the reviewer's concern regarding the amount of ground-truth validation. To address this, we substantially expanded the evaluation beyond disclosed models in Table 2 and include a **error comparision** experiment that systematically provides ground-truth validation for all estimates.
>
> We analyze every model on Hugging Face with valid disclosed emissions values. To avoid numerical instability and heavy-tailed distortion, we remove unreliable disclosures and retain the **middle 95%** of samples by relative error. The updated robust evaluation yields:
> | Metric | Value |
> |--------|-------|
> | Median RE | 0.32 |
> | MAPE | 0.42 |
> | Hit rate (×2 / ×3) | 0.74 / 0.82|
>
> Results show that **74–82% of predictions fall within 2–3×** of the reported emissions, with a median RE far below 1×. This demonstrates that our framework produces robust estimates under disclosure sparsity.

---

> ### Author Response · Authors · 2025-11-21
> **Response to Reviewer cMNg (3/4)**
>
> **W6**:"...It would strengthen the work to validate these approximation formulas using available disclosed training logs or verified benchmarks where possible...""
>
> **A6**: Thank you for this suggestion. We fully agree that validation against disclosed training logs is essential. In fact, our methodology already integrates this:
> - All models with verifiable training logs/electricity/emissions disclosures are categorized as Tier-1. These verifiable training logs are retrieved from published papers or technical reports;
> - These Tier-1 logs provide the ground-truth used to fit our regression, including FLOPs–GPU-hours relationships and hardware-specific scaling;
> - Our approximation formulas (Tier-2/Tier-3) are directly calibrated and evaluated using these Tier-1 ground-truth datapoints.
>
> Thus, our framework already performs the validation the reviewer proposes, using the complete set of publicly disclosed training logs available to date.
>
> **W7**: "The study provides limited ground-truth validation, relying on comparisons with only a small number of disclosed models (e.g., BLOOM, LLaMA) as shown in Table 2. While these examples demonstrate reasonable consistency, the validation set is too narrow to thoroughly evaluate the accuracy and robustness of the proposed estimation framework."
>
> **A7**: We thank the reviewer for raising this important point. In the revised version, we substantially expanded the validation beyond the original table. We now evaluate the estimation framework using all available models with reliable disclosures of training emissions or electricity use (**292 models, covering 5.58% of the Hugging Face models (dowloads>5k)**). Evaluation Results are shown in our response to W5  and Table 3 in the revised paper.
>
> Furthermore, to assess generalization under realistic missingness patterns, we added a **pseudo-missingness experiment** that constructs synthetic Tier 2 / Tier 3 samples from Tier 1 ground truth, as shown in responses to W4. Full details and tables are provided in Appendix A.4.
>
>
> **W8**:" Future work could expand the validation dataset by incorporating more models with partial disclosures, such as reported FLOPs or runtime statistics."
>
> **A8**: Our current framework already incorporates partially disclosed information, including reported FLOPs, GPU counts, and runtime statistics when available. These cases are included in our Tier-2 validation and are compared against our regression-based estimators.
>
> To facilitate future work in this direction, we will release all processed data (including models with full or partial disclosures), enabling the community to continually update and extend the validation set as new disclosures become available.
>
> **W9**:"...collaborating with model developers to access authentic training logs would provide stronger empirical grounding..."
>
> **A9**: We fully agree that direct collaboration with model developers would greatly strengthen empirical validation. In fact, this work is intentionally designed as the first foundational step toward such collaborations: by providing a consistent estimation pipeline and an openly accessible dataset, we aim to create the infrastructure that enables developers to contribute verified logs and cross-check our estimates.

---

> ### Author Response · Authors · 2025-11-21
> **Response to Reviewer cMNg (4/4)**
>
> **Q1**. "... provide more detailed comparison data—such as the disclosed FLOPs, GPU-hours, or runtime statistics—for these models..."
>
> **R1**: We thank the reviewer for the suggestion. We will release the full list of
> models that provide any training-related disclosures (emissions, electricity,
> runtime, GPU count, etc.) together with our dataset for full transparency. All disclosed values will be included in the released dataset. To show the availability of such data, we report the disclosure rates below:
> | **Metadata Field**              | **Count** | **Disclosure Rate** |
> |---------------------------------|-----------|----------------------|
> | Training emissions (tCO₂e)      | 292       | 5.58%               |
> | Electricity use (MWh)           | 15        | 0.29%               |
> | Grid emission factor            | 5         | 0.10%               |
> | Training region                 | 54        | 1.03%               |
> | GPU type                        | 955       | 18.25%              |
> | TPU Pod                         | 159       | 3.04%               |
> | Training runtime hours          | 179       | 3.42%               |
> | Training device count           | 414       | 7.91%               |
>
> **Table:** Metadata disclosure sparsity across the Hugging Face Models (downloads>5k).
>
>
> **Q2**. "... calculate and report the relative errors between the disclosed emissions and the framework’s predicted values..."
>
> **R2**: To address this, we include an error comparison experiment that systematically provides ground-truth validation for all estimates. We analyze every model on Hugging Face with valid disclosed emissions values. To avoid numerical instability and heavy-tailed distortion, we remove unreliable disclosures and retain the middle 95% of samples by relative error. Results show that **74–82% of predictions fall within 2–3×** of the reported emissions, with a median RE far below 1×. The evaluation results are shown in Table 3 of the revised paper.
>
> **Q3**. "...provide a quantitative breakdown of the overall emissions uncertainty, distinguishing the respective contributions from FLOPs estimation, hardware efficiency assumptions, and regional grid carbon intensity"
>
> **R3:** We agree with the reviewer and have added a quantitative decomposition of the overall emissions uncertainty. Using realistic perturbations to the three key components (FLOPs ±30%, hardware/runtime/PUE ±20%, and grid emission factor ±10%) and 1,000 Monte Carlo samples per model, we estimate each factor’s contribution to total variance.
>
> The averaged variance shares are:
> - **FLOPs estimation:** ~66%
> - **Hardware/runtime/PUE:** ~27%
> - **Grid emission factor:** ~7%
>
> This breakdown shows that FLOPs estimation is the dominant source of uncertainty, hardware assumptions contribute moderately, and regional carbon intensity accounts for only a small share. The requested quantitative analysis is included in Appendix A.3 of the revised manuscript.

---

> ### Comment · Reviewer_cMNg · 2025-11-21
> **Response for (1/4)**
>
> **For A1:**
> Thank you for taking the time to clarify this. Your explanation helps resolve most of my original concerns about novelty. In particular, the points you raised about treating the Hugging Face ecosystem as a meaningful unit of analysis, and about the practical challenges of building a unified large-scale estimation pipeline, make the contribution clearer to me. I appreciate the additional context.
>
> **For A2:**
> I previously highlighted the aspects that are similar to earlier work such as the motivation and the basic idea, but I agree with the authors’ explanation of how their method differs from prior approaches. Since the core of your framework is FLOPs-based attribution, I personally feel it would help the paper a lot if you added a short, focused paragraph that clearly explains how your FLOPs-based approach differs from earlier work. The papers [1–4] are only part of the broader landscape, and I think many readers would better understand the value of your contributions with a more explicit comparison.  In [1], Figure 1 explicitly charts training compute in petaflop/s-days across major models, which is essentially the same motivation that underlies your current study. Papers [2–4] also examine FLOP/s and hardware efficiency. Given this, I believe the paper would be strengthened by a direct paragraph explaining how your FLOPs-based attribution framework differs from these earlier FLOPs related analyses. Similarly, [5] also discusses compute energy CO₂ conversion and the challenges caused by missing metadata.
>
> It would be more helpful if this paper could more clearly distinguish its contributions from references [1-5] (such as the A2 response) and other relevant papers. A detailed comparison section would make the unique contributions of this paper clearer. In my opinion, this type of paper differs from other algorithm design papers; it should emphasis on analysis.

---

> > ### Comment · Reviewer_cMNg · 2025-11-21
> > **Response for (2/4)**
> >
> > **For A3:**
> > Faiz, Ahmad, et al. introduce LLMCarbon, which can also account for inference-related emissions. I understand that LLMCarbon requires more detailed metadata to complete the calculation, but I’m still curious whether there is any possibility of combining their approach with yours—at least for the Tier 1 models where detailed information is already disclosed. A brief discussion of how these two perspectives could complement each other would be helpful.
> >
> > - Faiz, Ahmad, et al. "Llmcarbon: Modeling the end-to-end carbon footprint of large language models." arXiv preprint arXiv:2309.14393 (2023).
> >
> > **For A4:**
> > Thank you for running the experiments. This is exactly the type of ablation I was hoping to see, and it clears up my concerns about the estimation error.
> >
> > **For A5:**
> > You mention that the evaluation was expanded beyond the models listed in Table 2 and that you analyzed every Hugging Face model with valid emissions disclosures. Could you provide a bit more detail about what those additional models are? Right now, only the aggregate results are shown, and it’s a little hard for me to understand what the expanded set actually includes.

---

> > > ### Comment · Reviewer_cMNg · 2025-11-21
> > > **Response for (3/4)**
> > >
> > > **For A6 & A8:**
> > > Thank you for clarifying this point. My earlier comment wasn’t referring to the usual Tier 1 or Tier 2 settings, but your explanation in A4 already addressed the concern I had, so I’m satisfied with that.
> > >
> > > **For A7:**
> > > Thank you for running the additional experiments. This resolves my concerns about the validation set.
> > >
> > > **For A9:**
> > > Thank you for the clarification.

---

> > > > ### Comment · Reviewer_cMNg · 2025-11-21
> > > > **Response for (4/4)**
> > > >
> > > > Thank you very much for your responses to Q1, Q2, and Q3. I don’t have any further concerns about these points. One small suggestion: it would be clearer if the ***revisions in the PDF were highlighted using different colors***, so readers can easily see what has been changed.

---

> > > > > ### Author Response · Authors · 2025-11-21
> > > > > **Response to the Reviewer’s Follow-up Questions**
> > > > >
> > > > > Thanks for your valuable advice and comments. We will make revisions and highlight them in the PDF. Below are our responses to the follow-up questions.
> > > > >
> > > > > > **Suggestion 1: It would be more helpful if this paper could more clearly distinguish its contributions from references [1-5] (such as the A2 response) and other relevant papers.**
> > > > >
> > > > > **Response**: In the revision, we will expand the literature review to contrast our framework with all related references in the same style as our A2 response to highlight differences in goals, assumptions, system boundaries, and applicability.
> > > > >
> > > > > > **Suggestion 2: whether there is any possibility of combining their approach with yours—at least for the Tier 1 models where detailed information is already disclosed. A brief discussion of how these two perspectives could complement each other would be helpful.**
> > > > >
> > > > > **Response**:Yes. While our main analysis focuses on training, our framework can be extended to inference. The inputs can switch to inference-specific quantities: the power and throughput of the inference hardware (often different from training GPUs), the efficiency and batching characteristics of inference workloads, and the compute required per generated token. Using these inputs, our framework can yield per-token inference-emission estimates in exactly the same way as for training. We will add a brief discussion of this extension in the revised manuscript.
> > > > >
> > > > > > **Suggestion 3: Could you provide a bit more detail about what those additional models are? Right now, only the aggregate results are shown, and it’s a little hard for me to understand what the expanded set actually includes.**
> > > > >
> > > > > **Response**: Beyond the examples already shown in Table 2 (Bloom, CodeLlama, Stable Diffusion, SAM, SAM2, BioCLIP, etc.), the expanded set of 292 models also includes：
> > > > > - Recent Meta Llama 3/3.1/3.2 and Llama-4 variants (e.g., meta-llama/Llama-3.1-405B, meta-llama/Llama-3.1-70B, Meta-Llama-3-70B),
> > > > > - AllenAI’s OLMo and OLMoE models (e.g., allenai/OLMo-7B-hf, allenai/OLMo-2-1124-13B-Instruct),
> > > > > - EleutherAI's gpt-neox-20b
> > > > > - Image and video models from Stability AI (stable-diffusion-2, stable-video-diffusion-img2vid and its variants).
> > > > > - A large cluster of biomedical language models from the OpenMed organization.
> > > > > - Miscellaneous models (e.g., ModernBERT variants, rerankers, and tiny classifiers)
> > > > >
> > > > > Overall, disclosed training emissions in this set range from around 10⁻³ tCO₂e for small fine-tuned models up to ≈8.9×10³ tCO₂e for frontier models such as Llama-3.1-405B.
> > > > >
> > > > > In the revision, we will add a summary in Appendix A.3 that groups these models.

---

> > > > > > ### Comment · Reviewer_cMNg · 2025-11-21
> > > > > > **New Response**
> > > > > >
> > > > > > Thank you again for your detailed responses. I don’t have any further concerns at this point. I’ll update my score once the revised PDF is uploaded.

---

> > > > > > > ### Author Response · Authors · 2025-11-23
> > > > > > > **Revised Manuscript Uploaded**
> > > > > > >
> > > > > > > Thank you very much for your positive feedback. We have incorporated the requested revisions, highlighted in blue in the uploaded PDF. You may refer to it for details. We sincerely appreciate your time and review.

---

### Author Response · Authors · 2025-11-28
**General Response to the Area Chair and Reviewers**

Dear Area Chair and Reviewers,

Thank you for your time and thoughtful evaluations during the review and rebuttal. We are grateful to the reviewers for recognizing the contribution of our work to the community. We appreciate the opportunity and would like to briefly clarify why we believe the paper makes a meaningful and original contribution.

> **Clarification on the Novelty of Our Work**

- A fundamental question is the scale of carbon emissions generated by the AI industry. Until now, the community has lacked any empirical corpus large enough to enable systematic quantification. Our work identifies an unexplored opportunity. The Hugging Face platform can be treated as a large scale, publicly accessible, and audit ready corpus for carbon accounting. This perspective does not appear in prior literature. It provides the community with a new empirical foundation and enables, for the first time, an aggregated view of emissions across thousands of open models.

- On top of this empirical foundation, we introduce a unified estimation pipeline designed to operate across heterogeneous metadata, inconsistent logging, and incomplete disclosures. This capability is important for making large-scale carbon accounting possible and goes beyond routine engineering. It provides the analytical framework required to make previously unquantifiable parts of the platform measurable.

- Regarding accuracy, our objective is not to reconstruct exact per-model footprints, which is infeasible under current disclosure practices, but to produce conceptually sound and quantitatively reliable estimates with uncertainty well below an order of magnitude. This is precisely the level needed to resolve the scale question the community currently cannot answer: whether the open-model ecosystem emits thousands, tens of thousands, or hundreds of thousands of tons of CO₂e. Our results provide the first concrete answer at the correct scale. While perfect precision is impossible, we emphasize that we made every feasible effort to ensure the estimates are accurate and well grounded given the available disclosures and quantified uncertainty bounds.

Since the submission, we have substantially expanded and refined the dataset under the same estimation framework. We collected additional information from model cards, official websites, technical reports, published papers, GitHub discussions, and other related sources, all manually checked for correctness. These additions allow more precise classification, and resolve ambiguous cases. As a result, our dataset grew from roughly two thousand to over five thousand models, yielding reliable conclusions under the same estimation logic.

> **Revisions in the Updated Manuscript (highlighted in blue)**

In the revised manuscript we have uploaded, we implemented several updates that aimed at addressing the reviewers’ concerns, including:

1. We added a full error-comparison experiment using models with verifiable emissions disclosures, together with filtering to remove noisy or inconsistent logs.

2. We added an uncertainty-propagation section (Eq. 7 and Appendix A.2) to match precision with uncertainty bounds. We revised all numerical reporting to follow significant-digit rules and aligned the precision of reported totals with their propagated uncertainty.

3. We introduced a synthetic Tier-2/Tier-3 evaluation (pseudo experiment) by masking Tier-1 metadata to emulate realistic missingness, enabling quantitative assessment of Tier-2/Tier-3 estimation errors where ground truth is unavailable.

4. To quantify uncertainty among different inputs, we incorporated a variance analysis, separating the effects of FLOPs estimation, hardware/runtime/PUE assumptions, and regional emission factors.

5. We expanded the related-work section to include LLMCarbon, CodeCarbon, and CarbonTracker, and added a methodological comparison clarifying the novelty of our framework.

6. We made several focused clarifications and minor textual revisions to directly address the remaining reviewer comments.

We hope these clarifications help convey the significance of the contribution, and we believe our work can provide a scalable empirical foundation for carbon accounting of open-source AI training. If there are any further questions, we would be glad to clarify or discuss them further during the remaining rebuttal period.

Sincerely,

The Authors

---

### Note · Program_Chairs · 2026-01-17
**Submission Desk Rejected by Program Chairs**

The following references in this submission do not refer to real documents and/or have major errors in bibliographic information:

 "Michael Percy, Alison Kennedy, et al. Carbon connect: An ecosystem for sustainable computing. arXiv preprint arXiv:2405.13858, 2024."